

# Droplet activation behaviour of atmospheric black carbon particles in fog as a function of their size and mixing state

Ghislain Motos[1], Julia Schmale[1], Joel Christopher Corbin[1,*], Marco Zanatta[1,**], Urs Baltensperger[1] and Martin Gysel[1]

[1]Laboratory of Atmospheric Chemistry, Paul Scherrer Institute, 5232 Villigen PSI, Switzerland
[*]Now at Measurement Science and Standards, National Research Council Canada, 1200 Montreal Road, Ottawa K1A 0R6, Canada
[**]Now at Alfred Wegener Institute, Helmholtz Centre for Polar and Marine Research, Bremerhaven, Germany

*Correspondence to*: Martin Gysel (martin.gysel@psi.ch)

**Abstract.** Among the variety of particle types present in the atmosphere, black carbon (BC), emitted by combustion processes, is uniquely associated with harmful effects to the human body and substantial radiative forcing of the Earth. Pure BC is known to be non-hygroscopic, but its ability to acquire a coating of hygroscopic organic and inorganic material leads to increased hygroscopicity as well as diameter, facilitating droplet activation. This affects BC radiative forcing through aerosol-cloud interactions (aci) and BC life cycle. To gain insights into these processes, we performed a field campaign in winter 2015/16 in a residential area of Zurich which aimed at distinguishing different particle mixing states regarding hygroscopic properties in the cloud condensation nuclei (CCN)-activated fraction spectrum of urban aerosol and establishing relations between the mixing state of BC and its activation to form droplets in fog. This was achieved by operating a CCN counter (CCNC), a scanning mobility particle sizer (SMPS), a single particle soot photometer (SP2) and an aerosol chemical speciation monitor (ACSM) behind a combination of a total- and an interstitial-aerosol inlet.

Our results indicate that, depending on the time of the day, we sampled both heavily aged internally mixed BC from background air advected to the site and freshly emitted externally mixed BC from local or regional traffic sources. During rush hours in the morning of weekdays, we found clear evidence that the enhanced traffic emissions caused peaks in the number fraction of externally mixed BC particles which do not act as CCN within the CCNC. The mixing state of BC particles was also found to play a key role in their ability to form fog droplets. The very low effective peak supersaturations ($SS_{peak}$) occurring in fog (between approximately 0.03 and 0.06% during this campaign) restrict droplet activation to a minor fraction of the aerosol burden (around 0.5 to 1% of total particle number concentration between 20 and 593 nm) leading to very selective criteria on diameter and chemical composition. We show that bare BC cores are unable to activate to fog droplets at such low $SS_{peak}$, while BC particles surrounded by thick coating have very similar activation behaviour as BC-free particles. The threshold coating thickness required for activation was shown to decrease with increasing BC core size. Using simplified $\kappa$-Köhler theory combined with the ZSR mixing rule assuming spherical core-shell particle geometry constrained with single particle measurements of respective volumes, we found good agreement between the predicted and the directly observed size and mixing state resolved droplet activation behaviour of BC-containing particles in fog. This successful closure demonstrates the predictability of their droplet activation in fog with a simplified theoretical model only requiring size and mixing state information, which can also be applied in a consistent manner in model simulations.



## 1 Introduction

Black carbon (BC) is formed during the incomplete combustion of fossil and biogenic fuels in anthropogenic sources (e.g. on-road and off-road diesel vehicles, residential heating) and natural sources (natural wildfires and smoldering peat fires). According to a recent study based on emission inventory modelling (Klimont et al., 2017), 75% of the global atmospheric BC mass in the year 2010 originated from human activities. It has to be noted that the authors of this study did not perform any formal uncertainty analysis. Although BC represents a small fraction of the atmospheric particulate matter (typically around 10% by mass; Putaud et al., 2004 and Lanz et al., 2010 over Europe and Hueglin et al., 2005 in Switzerland), it possesses unique properties that lead to strong impacts on health and climate. Indeed, sufficient evidence has now been brought to link exposure to BC with cardiopulmonary morbidity and mortality (World Health Organization, 2012). Concerning the impacts on climate, BC has been shown to influence the Earth's climate *via* both aerosol-radiation interactions (ari, industrial-era forcing of +0.71 W m$^{-2}$, 90% uncertainty range: +0.08 to +1.27 W m$^{-2}$; Bond et al., 2013) and aerosol-cloud interactions (aci, industrial-era forcing of +0.23 W m$^{-2}$, 90% uncertainty range: -0.47 to +1.0 W m$^{-2}$; Bond et al., 2013). The high uncertainties attached to these estimates originate from the low level of confidence in understanding and quantifying the atmospheric processes in which BC is involved, particularly with respect to aerosol-cloud interactions (Bond et al., 2013). The internal mixing of BC with other material is a key factor affecting its radiative forcing, since it has impacts on both ari and aci of BC. Such internal mixing focuses incident solar radiation to the BC core and results in an increase of its mass-specific absorption (Bond et al., 2006; Lund et al., 2017; Cappa et al., 2012). The atmospheric lifetime of BC is also influenced by its mixing state through nucleation scavenging (Lund et al., 2017). This is of major importance, as an increased lifetime allows for interactions with the solar radiation during a longer time window (Hodnebrog et al., 2014). Lund et al. (2017) modelled the changes of global mean ari-induced radiative forcing (RFari) when varying the amount of coating required for a particle to pass from the non-hygroscopic mode (unactivated) to the hygroscopic mode (activated to a droplet). They reported changes up to 25-50% of the RFari compared to the baseline simulation. It is therefore of major importance to better assess the dependence of the BC activation behaviour on its size and mixing state.

General definitions of fog include two criteria for suspended water droplets to be called fog: a vicinity to the Earth's surface and a reduction of visibility below 1 km (e.g. American Meteorological Society, Glickman, 2000; National Oceanic and Atmospheric Administration, NOAA, 1995). Fog is a type of cloud which forms upon isobaric processes. The detailed microphysics associated with these processes can be found in Pruppacher and Klett (1980). On a global scale, fog is relatively sparse (total amount of 1% over both sea and land; Warren et al., 2015) but its spatial coverage is highly variable around the globe, up to an amount of 40% (e.g. Gordon et al., 1994; Lange et al., 2003; Syed et al., 2012).

Depending on the process of formation, different types of fog can be distinguished: radiation fog, advection fog, sea and steam fog, mixing fog and ice fog. The most common type is radiation fog, which is formed by isobaric infrared cooling of the Earth's surface. The air in contact with the surface is then cooled by conduction, decreasing the temperature of the humid boundary layer air by atmospheric mixing. If the dew point temperature of the air mass is reached, fog forms. The required meteorological conditions are clear skies and wind speed below 0.5-1 m s$^{-1}$ (Roach et al., 1976; Mason, 1982). Therefore, radiation fog generally occurs after sunset, but can persist all day in winter, if not dissipated by solar radiation. Several field studies have been performed to





investigate the physical processes of fog formation and dynamics (Haeffelin et al., 2010) and the evolution of chemical species in the presence of fog (Fuzzi et al., 1992). The cooling of an air parcel below its dew point results in the supersaturation (SS) of water vapour. Droplet activation of an aerosol particle occurs when the SS of the surrounding water vapour exceeds its critical supersaturation ($SS_{crit}$), thereby forming a cloud or a fog

droplet.

BC is most often emitted bare or mixed with only small amounts of other materials; at this stage it hardly undergoes hygroscopic growth at elevated relative humidity (RH) because BC is water insoluble (Weingartner et al., 1997; Gysel et al., 2003). Several recent chamber and field studies showed that the subsequent acquisition of water-soluble coatings, by condensation and coagulation of organic and inorganic materials, enhances the

hygroscopicity of these BC-containing particles and allows droplet activation at atmospherically relevant SS (e.g. Tritscher et al., 2011; Liu et al., 2013; Wittbom et al., 2014). The fate of BC particles in fog has also been studied, mostly by comparing scavenging efficiencies of BC with other species. Hallberg et al. (1992), Noone et al. (1992) and Facchini et al. (1999) showed that elemental carbon (EC) is preferentially found in interstitial particles rather than in fog droplets, while Gundel et al. (1994) found evidence supporting the hypothesis that

organic compounds could enhance the incorporation of BC into fog droplets. Results from Collett et al. (2008) indicate that the scavenged fraction of BC is higher for wood smoke emissions than for vehicle exhaust emissions. A single-particle analysis of BC in low-altitude stratocumulus clouds, in which low $SS_{peak}$ were retrieved, showed that the activation of BC was made possible by the presence of coatings (Schroder et al., 2015). However, the technical complexity of such measurements did not yet allow for a precise quantification of

the activation behaviour of BC as function of its size and coating thickness ($\Delta_{coating}$).

Few model studies have represented the role of BC in aerosol-cloud interactions (Bond et al., 2013). To simulate the cloud properties of ambient particles, the increase of hygroscopicity of BC has to be accurately represented, meaning that the models need realistic mixing state schemes. Due to the scarcity of instruments that can provide this type of information and to high computational costs, these properties are often modelled in a highly

simplified manner. The conversion from hydrophobic to hydrophilic BC (which may lead to droplet activation) was originally considered to happen after a fixed lifetime (Koch et al., 2009). This conversion has recently been treated as a variable depending on e.g. particle concentration in many particle-resolved models (e.g. Riemer et al., 2009). The results from these recent simulations emphasized the importance of accurately simulating the increase of BC hygroscopicity with aging in order to get realistic assessments of the corresponding

concentrations and radiative forcing, with crucial implications for specific research questions such as the estimation of the climate impact of BC in highly polluted regions (e.g. eastern Asia; Matsui, 2016) or the transport of BC to the Arctic (Liu et al., 2011).

Significant efforts are needed to reach a better understanding of the evolution of the mixing state of BC after emission, and quantify the links between mixing state and droplet activation. Chamber measurements recently

started to address this question (e.g. Dalirian et al., 2018) but very few studies report ambient measurements. Urban areas contain a variety of BC sources, making them favorable sites to study different mixing states of BC. Furthermore, the occurrence and stability of fog at ground level in these areas facilitates the study of the activation behaviour of BC. In this study, we first focus on the size-dependent mixing state and hygroscopicity of aerosol particles emitted in winter at an urban site, before establishing quantitative links between particle

diameter, mixing state and droplet activation of BC-containing particles. Then, we estimate the $SS_{crit}$ of BC-



containing particles using a theoretical approach based on a core-shell model and compare the predicted activation behaviour with in-situ field measurements of droplet activation in fog. We found agreement between predicted droplet activation of BC, constrained with measured particle size and BC volume fraction, and observed droplet activation in the fog. This finding justifies the simplified description of BC activation in model

simulations based on particle size and BC volume fraction using $\kappa$-Köhler theory.

## 2 Measurements and methods

### 2.1 Measurement period and site

The field campaign took place at the Irchel campus of the University of Zurich, located 2.5 km north of the city center (47°23'43" N, 8°32'55" E) during winter 2015/16. The data presented here come from measurements

performed over the period from 6 November 2015 to 31 January 2016. The instruments used for this campaign ran in an 11.2 m$^3$ air-conditioned stationary trailer.

The Irchel campus is located within a residential area of Zurich; the closest industries or agricultural fields are located 2 km away from the measurement site. One of the most used highways in Switzerland passes eastward and northward of the measurement site, the closest point being 2.5 km northeast (96'877 to 142'074 car counts

in total in December 2015, depending on the exact location (source: SARTC). Smaller busy roads are found around 200 m northward and westward of the site. In addition, wood burning emissions from domestic heating are also expected to contribute to the anthropogenic aerosol loading at this location during winter time.

The Swiss plateau is known for a high frequency of fog events occurring during winter. For example, during the period 1901-2012, continuous fog or low stratus presence during a full 24-hour period was observed on average

17 days in total in Zurich in the months from September to March (28 days with at least half-day occurrence; Scherrer and Appenzeller, 2014). Thus, due to the high frequency of foggy conditions and the presence of mixed sources, this measurement site was chosen.

### 2.2 Instrumentation

Two different inlets and twelve instruments were used during this campaign (Fig. 1). All aerosol particles,

including fog droplets, were sampled through a hood-shaped total inlet with a flow rate of 4.8 L min$^{-1}$, approximately 3 m above the ground. One meter away at the same height, an interstitial inlet sampled non-activated particles with a flow rate of approximatively 16.7 L min$^{-1}$. This inlet included an aerodynamic size discriminator removing all large particles and hydrometeors (Very Sharp Cut Cyclone, BGI, Butler, NJ, USA; described in Kenny et al., 2000). Laboratory tests showed small variations of the cut-off diameter (2.2 to

2.4 μm) for flow rates between 15.7 and 17.7 L min$^{-1}$. This range of cut-off is close to the value of 2.6 μm recommended by Hammer et al. (2014b) for separating hydrated (but non-activated) particles from fog droplets. Setting the cut-off between the diameter modes of non-activated (but hydrated) particles and fog droplets is very important for obtaining reliable results. If it is set too high, activated droplets may enter the interstitial line and the resulting curve of the size-dependent activated fraction of particles gets flattened; if it is set too low, non-

activated but large solution droplets may be removed by the inlet, resulting in an artificially increased activated fraction. Due to the different particle losses in the interstitial and the total lines, scaling factors were calculated using the ratios of the total to the interstitial particle number size distributions over fog-free, sunny periods,



during which these size distributions should be identical below the interstitial inlet cut-off diameter. For each fog event, scaling factors were calculated before and after the event, averaged, and then used during the event to correct the particle number size distribution behind the interstitial inlet. For the single-particle soot photometer (SP2), a scaling factor of 1.16, independent of particle diameter, was used until 17 December (on that day, a thin tubing causing a pressure drop to the SP2 only was replaced by a thicker one; after that day, the measured scaling factor was 1.03). For the scanning mobility particle sizer instruments, size-dependent scaling factors were calculated for each fog event in order to take into account both the different line losses behind each inlet and the internal measurement errors of each SMPS.

Aerosols from both inlets were then led inside the trailer by stainless steel tubes and dried with vertically positioned diffusion driers, before being brought to the instruments with electrically conductive tubing. These driers were needed to keep the relative humidity below 40% inside the measurement lines, as recommended by the World Meteorological Organization/Global Atmosphere Watch (WMO/GAW, 2016).

### 2.2.1 Scanning cloud condensation nuclei number and sCCNC-activated fractions

In order to get size-dependent information on the hygroscopic properties of ambient particles, a scanning cloud condensation nuclei counter (sCCNC) sampled air behind the total inlet. The sCCNC consists of a differential mobility analyzer (DMA, model TSI long, TSI Inc., Shoreview, MN, USA) scanning the particle mobility diameter range from 20 to 593 nm in 5.5 min, the resulting monodisperse aerosol is split between a CCNC (model CCN-100, Droplet Measurement Technologies, Longmont, CO, USA; Roberts and Nenes, 2005) and a condensation particle counter (CPC model 3022, TSI Inc., Shoreview, MN, USA). The assembly DMA-CPC can also be used as a scanning mobility particle sizer (SMPS) and provides the particle number size distribution behind the total inlet. The CCNC changes SS every 11 minutes, covering nine SS: 0.14, 0.21, 0.27, 0.34, 0.40, 0.47, .67, 0.93 and 1.33%. Scans with unstable temperature in the CCNC chamber were removed from the analysis. The sCCNC, which was presented by Moore et al. (2010), was e.g. used in a semi-urban environment by Jurányi et al. (2013) and in a boreal forest by Paramonov et al. (2013) and permits the comparison of particle (N) number size distribution and CCN number size distribution with a time resolution of 5.5 min for a fixed SS. Activated fractions were calculated from these two size distributions after correcting both measurements for multiple charging. They are referred to as sCCNC-activated fractions.

### 2.2.2 Particle number and size distribution

A second SMPS, which combined the same models of DMA and CPC as the one sampling behind the total inlet, was used behind the total inlet but scanned over a larger mobility diameter range from 19 nm to 807 nm. The comparison of particle number size distributions behind the total and interstitial inlets allows for the calculation of the dry activation cut-off diameter, as explained in Sect. 2.3.3, which corresponds to the ambient SS present when the fog formed.

### 2.2.3 Black carbon

A single particle soot photometer (SP2, Droplet Measurement Technologies, Longmont, CO, USA) upgraded to 8-channel Revision C version was the only instrument switching between the total and interstitial inlet, through an automated three-way valve, with a 20 min alternation. Detailed information about the SP2 can be found in



Moteki and Kondo (2007); Schwarz et al. (2006) or Stephens et al. (2003). Briefly, the SP2 carries the aerosol sample flow (0.12 L min$^{-1}$) through a high-intensity intra-cavity Nd:YAG laser with a wavelength of 1064 nm, making BC particles incandesce (detection by two photomultipliers) until they vapourize. An avalanche photodiode is used to detect elastically scattered light. A second multi-photodiode was used as a split detector,

providing information on the position of particles in the laser beam (Gao et al., 2007). The peak intensity of the thermal radiation is proportional to the refractory BC (rBC) mass in the particles, from which the rBC mass equivalent diameter ($D_{rBC}$) is inferred assuming spherical shape. The peak amplitude of the elastically scattered light is used for optical sizing of BC-free particles. The SP2 was calibrated before and after the campaign using mobility diameter selected fullerene soot for rBC mass (mobility-mass relationship taken from Gysel et al.,

2011) and polystyrene latex spheres (PSL, 269 nm) for the scattering detector. Calibrated scattering cross section measurements of BC-free particles were converted to optical diameters ($D_{opt}$) assuming spherical particles with a refractive index of 1.50+0i.

The presence of different types of detectors in the SP2 provides an opportunity to obtain information on the BC mixing state on single particle level. When an internally mixed BC-containing particle enters the laser beam, it

heats up and the coating evaporates resulting in a reduction of the scattering cross section, followed by further heating of the remaining BC core until the BC boiling point is reached and the BC core starts evaporating. The peak incandescent signal occurs when the BC boiling point is reached. As laser intensity increases and scattering cross section decreases when the particle enters the laser beam, the peak scattering signal can either occur a few microseconds before peak incandescence when coating evaporation begins or at peak incandescence when BC

core evaporation begins. The time difference between scattering and incandescence peak signals, commonly referred to as delay time method, can be used for a binary distinction between BC particles with thick coatings (>70% coating by volume according to unpublished data from our laboratory) and BC particles with moderate or no coating at all (Moteki et al., 2007).

A second, more quantitative method to determine $\Delta_{coating}$ was proposed by Gao et al. (2007): the leading-edge-

only (LEO)-fit. As BC-containing particles evaporate due to strong heating, their scattering cross section is less than the original value by the time they reach the centre of the laser beam where the peak scattering signal would occur for BC-free (i.e. non-evaporating) particles, thus disqualifying measured scattering amplitude for optical sizing. However, knowing the particle position in the laser beam from the split detector signal makes it possible to use the unperturbed leading edge scattering signal, i.e. before evaporation onset, for particle optical

sizing. Scattering cross sections measured for BC-containing particles were converted to $D_{opt}$ assuming a coated sphere morphology with BC core volume constrained from the rBC mass measurement and assuming refractive indices of 2.00+1.00i and 1.50+0i for BC core and coating, respectively. Details of the data analysis approach are provided in Laborde et al. 2012a, 2012b. By subtracting the rBC mass equivalent core radius from the optical radius of the unperturbed particle, we obtain $\Delta_{coating}$. For the data analysis of the present work, we used

the leading edge scattering signal at 3% of the maximal laser intensity.

An aethalometer (model AE 33, Magee Scientific, Berkeley, CA, USA) was placed behind the total inlet. This instrument measures the attenuation of light, at seven different wavelengths from 370 to 950 nm, passing through a filter that gets continuously loaded with ambient aerosols. The near-infrared channel at a wavelength of 880 nm was used for extracting the equivalent black carbon (eBC) mass concentration from the measured

attenuation coefficient (e.g. Weingartner et al., 2003). The eBC mass concentrations reported by the instrument



firmware were used without adjustment (i.e. default mass attenuation coefficient and no loading compensation). The term "eBC" is used following the recommendation by Petzold et al. (2013) in order to express that the accuracy of the inferred eBC mass concentration depends on the accuracy of the measured attenuation coefficient (e.g. shadowing effects) and accuracy of the mass attenuation cross section assumed to convert from

the attenuation coefficient to the eBC mass concentration. The spectral dependence of the aerosol light absorption, commonly expressed with the absorption Ångström exponent (AAE), which we determined from the aethalometer measurements at 470 and 880 nm. The AAE calculated in this manner can be used for black carbon source apportionment, if traffic and wood burning are its only main sources and if the AAE of either source is well known (Zotter et al., 2017, and references therein).

**2.2.4 Aerosol chemical composition**

To get information on the chemical composition and the mass of the non-refractory submicron bulk aerosol, a time-of-flight aerosol chemical speciation monitor (ToF-ACSM; Fröhlich et al., 2013), an instrument based on the aerosol mass spectrometer technology (AMS, Aerodyne Research Inc., Billerica, MA, USA), sampled air behind the total inlet. Six calibrations were performed, including pre and post campaign, and standard data

analysis procedures using the Tofwerk "IgorDAQ" software package (Tofwerk AG, Thun, BE, Switzerland) were applied (Fröhlich et al., 2013).

**2.2.5 Cloud microphysics**

Three instruments were installed on the roof of the trailer, approximately 3 m above the ground: a dew point mirror, a particulate volume monitor and a meteorological station. The dew point mirror (DPM; mirror-type dew

point hygrometer VTP37 Airport, Meteolabor AG, Wetzikon, Switzerland) provided relative humidity data with a resolution of 0.1% by measuring both the ambient temperature and the dew point temperature. This instrument is designed to measure the dewpoint corresponding to the total condensed and gaseous water content. Accordingly, it indicates the presence of fog when the dew point exceeds ambient temperature due to the presence of liquid water. A particulate volume monitor (PVM; Gerber, 1991), which detects the light scattering

by the fog droplets in forward direction, provided a second independent measurement of the liquid water content (LWC). A meteorology station provided data of temperature, pressure, wind speed and direction, precipitation rate and solar flux.

**2.3 Data analysis and theory**

**2.3.1 Fog type and definition of a fog event**

The PVM and DPM were used to indicate the presence of fog. Visibility was not measured during this campaign. The LWC derived from PVM and the DPM measurements agreed within ±25% during the campaign. We used a minimum LWC of 1 g m$^{-3}$ measured by the PVM during at least one hour as threshold to define fog events. Throughout the field campaign, four fog events were retained in the analysis of the present study, all of them between 14 and 20 December 2015 (Table 2). They occurred during night time principally (see Table 2)

with low wind speed (Fig. 4). Thus, even though no classification of fog types was carried out during the campaign, it is highly probable that we only experienced radiation fogs. Other events were either too short, discontinuous, or suffered from a lack of instrumental data.





### 2.3.2 κ-Köhler theory and ZSR rule

The Köhler theory (Köhler, 1936) combines the Kelvin and Raoult effects to describe the equilibrium saturation vapor pressure ($RH_{eq}$) over a solution droplet. In the framework of the present study, this theory is the base for various calculations establishing a relationship between particle dry diameter ($D_{dry}$), chemical composition and

$SS_{crit}$ for CCN activation. Petters and Kreidenweis (2007) proposed a simple semi-empirical parameterization of the Raoult effect in which the $\kappa$ value is the single free parameter to describe particle hygroscopicity. The equilibrium supersaturation over the solution can then be expressed as:

$$SS_{eq}(D) := RH_{eq}(D) - 1 = \frac{D^3 - D_{dry}^3}{D^3 - D_{dry}^3(1-\kappa)} \exp\left(\frac{4.\sigma_{s/a}M_w}{RT\rho_w D}\right) - 1 \qquad (1)$$

Where $D$ is the solution droplet diameter, $D_{dry}$ is the dry particle diameter, $\sigma_{s/a}$ is the surface tension of the

solution-air interface (considered as pure water in our calculations), $M_w$ and $\rho_w$ are the molar mass and the density of water, respectively, $R$ is the universal gas constant and $T$ is the absolute temperature.

Knowledge of two out of three parameters in the relationship $SS_{crit}$-$D_{dry}$-κ allows calculating the third component by numerically solving Eq. (1). We made use of this relationship to infer the $SS_{crit}$ of individual BC-free as well as of BC-containing particles from their dry size and the κ value determined with the SP2 and other

instruments (Fig. 2). The SP2 provides a measurement of both rBC core mass equivalent diameter ($D_{rBC}$) and particle optical diameter ($D_{opt}$), which makes it possible to calculate the BC volume fraction ($\varepsilon_{rBC}$) for each particle falling within the relevant detection limits:

$$\varepsilon_{rBC} = \frac{D_{rBC}^3}{D_{opt}^3} = \frac{D_{rBC}^3}{(D_{rBC} + 2\Delta_{coating})^3} \qquad (2)$$

The optical particle diameter can also be expressed with $D_{rBC}$ and $\Delta_{coating}$. The rBC volume fraction is required to

calculate $\kappa_{mix}$, the κ value of internally mixed particles, which is equal to the volume fraction weighted mean of the κ values of all species or component classes present in the particle (Petters and Kreidenweis, 2007), under the assumption that the Zdanovski-Stokes-Robinson (ZSR) mixing rule (Stokes and Robinson, 1966) applies for the hygroscopic growth. We treat our particles as two component mixtures with considering an insoluble BC core ($\kappa = 0$) and a soluble coating to which we assigned the median κ value ($\kappa_{coating} := \kappa_{median}$) retrieved from

sCCNC measurements (see Sect. 2.3.4) of all particles of equal size. $\kappa_{mix}$ then becomes:

$$\kappa_{mix} = \kappa_{coating}(1 - \varepsilon_{rBC}) = \kappa_{median}(1 - \varepsilon_{rBC}) \qquad (3)$$

Combining Eq. (1) and (3) makes it possible to estimate $SS_{crit}$ of individual BC-containing particles by applying Köhler theory and ZSR rule to SP2 and sCCNC data providing particle size ($D_{rBC}$ and $\Delta_{coating}$), BC volume fraction $\varepsilon_{rBC}$ and coating hygroscopicity ($\kappa_{coating}$). These calculations are simplified in so far as spherical core-

shell morphology is assumed for inferring the particle optical diameter from SP2 raw signals and for the κ-Köhler theory.

### 2.3.3 Retrieval of activation cut-off diameters in fog

The size-resolved activated fraction is generally defined as the number fraction of particles at a certain $D_{dry}$ that formed an activated droplet. The combination of total and interstitial inlets in fog makes it possible to assess the

activation of the ambient aerosol to fog droplets (Hammer et al., 2014b): under the assumption that only fog


droplets were removed by the interstitial inlet, the difference between the total and interstitial particle number size distribution reflects the dry size distribution of particles that were activated. Dividing the dry size distribution of activated particles by the total size distribution provides the size-resolved activated fraction spectrum. To emphasize that this activated fraction results from instruments which measure atmospheric

activation, we refer to fog-activated fraction. By contrast, we use the terms sCCNC-activated fraction and to refer to the *potential* activation measured at controlled SS in the sCCNC. The 50% activation cut-off diameter ($D_{50}^{fog}$) is defined as the dry particle diameter at which the fog-activated fraction reaches 50%, whereas the half-rise activation diameter ($D_{half}^{fog}$) is defined as the diameter at which half of the maximum fog-activated fraction is reached. If the activation plateau at large particle diameters levels off at a maximum fog-activated fraction of

100%, then $D_{half}^{fog}$ equals $D_{50}^{fog}$.

Activated fractions were independently calculated using two different types of particle number size distribution measurements behind each inlet: from the SMPS and from the SP2. Results from both types of instruments agreed in general and showed distinct fog droplet activation at the largest particle diameters, while smaller particles remained interstitial. However, the signal-to-noise ratio in the fog-activated fraction spectra from the

SMPS measurements was poorer than for the SP2-derived fog-activated fraction spectra. For this reason, the values of $D_{50}^{fog}$ and $D_{half}^{fog}$ mentioned hereafter are extracted from the fog-activated fractions calculated with the SP2 number size distributions, specifically considering BC-free particles only (Fig. 3a). As the fog events lasted at least 3.5 hours each (Table 2), the average particle number size distributions measured over an entire event have a high statistical significance.

**2.3.4 Retrieval of $\kappa_{median}$ value from sCCNC measurements**

Using the sCCNC setup (Fig. 1) provides simultaneous measurements of the CCN as well as total particle number size distributions, and dividing the former by the latter results in the CCN-activated fraction spectrum (Fig. 3b). The diameter at which CCN activation reaches 50% for the SS applied in the sCCNC is commonly defined as the sCCNC-critical activation diameter, $D_{crit}^{sCCNC}$. Below, we will also refer to the diameter at which

CCN activation reaches 25% and 75% as $D_{25}^{sCCNC}$ and $D_{75}^{sCCNC}$. The median $\kappa$ value, $\kappa_{median}$, for particles with dry diameter $D_{dry} = D_{crit}^{sCCNC}$ of the sampled aerosol is calculated from measured $D_{crit}^{sCCNC}$ by considering the SS applied in the CCNC and using $\kappa$-Köhler theory (Eq. (1); surface tension of the droplets is assumed to be equal to that of water and the temperature at activation is assumed to be equal to the sample flow temperature in the CCNC). As an example, Figure 3c shows all $\kappa_{median}$ observed during the 14 December fog event as a function of $D_{crit}^{sCCNC}$.

Note, all $\kappa_{median}$ inferred from measurements at identical SS fall on a common line rather than being randomly scattered because $\kappa_{median}$ and $D_{crit}^{sCCNC}$ are unambiguously related through $\kappa$-Köhler theory for constant SS.

Observed $D_{crit}^{sCCNC}$ varied from < 20 nm to > 200 nm due to applying different SS and due to temporal variations in the aerosol hygroscopicity (Fig. 3c). Reaching larger $D_{crit}^{sCCNC}$ was not possible because the CCNC can only measure at SS greater than ~0.1%. As fog formation occurs at lower SS, knowledge of the $\kappa$ value for

$D_{crit}^{sCCNC}$ around 300 to 500 nm is required for interpreting the fog observations. Therefore, we extrapolate the size-resolved $\kappa_{median}$ data to $D_{half}^{fog}$ and $D_{50}^{fog}$, which are the estimated activation cut-off diameters for fog droplet formation, as illustrated in Figure 3c.





**2.3.5 Retrieval of effective peak supersaturation in fog**

The highest SS encountered by the activated particles in the fog during a sufficiently long time, which made them grow across their fog-critical diameter for becoming a stable cloud or fog droplet, is defined as the effective peak supersaturation ($SS_{peak}$; Hammer et al., 2014a). We use the $SS_{peak}$ when indirectly inferring it by

comparing observed dry particle cut-off diameter for droplet activation in the fog with the $D_{dry}$ to $SS_{crit}$ relationship from CCN counter (CCNC) measurements.

Inferring $SS_{peak}$ during a fog event is made possible by combining the value of the activation diameters (Sect. 2.3.3) and the hygroscopicity of particles activated to fog droplets (Sect. 2.3.4), using the κ-Köhler theory (Sect. 2.3.2). Two different values of $SS_{peak}$ corresponding to the two different activation diameters were calculated, as

it is unknown which one of the two diameters is closer to the actual cut-off (lacking measurements in the size range where the activation plateau is reached). The temperature at which particles activate was assumed to be the measured ambient temperature at 3 m above the ground.

**3 Results and discussion**

**3.1 Overview of particle concentration and hygroscopicity results**

Continuous measurements of particle and species concentrations from 6 November 2015 to 31 January 2016 gave the opportunity to observe the type of aerosol present at the campaign site in winter (see Table 1). The median total particle number concentration was 5'879 cm$^{-3}$ (interquartile range (IQR) = 5'967 cm$^{-3}$), with lower concentrations at night which often dropped below 2'000 cm$^{-3}$, and peaks reaching more than 20'000 cm$^{-3}$ for more than an hour in the morning rush hour period. The particle number size distribution was generally

unimodal, centered between 40 and 120 nm. The median eBC mass concentration was 1.1 μg m$^{-3}$ (IQR = 1.3 μg m$^{-3}$), with higher and more variable values during weekdays (1.3 μg m$^{-3}$; IQR = 1.4 μg m$^{-3}$) than weekend days (0.8 μg m$^{-3}$; IQR = 1.0 μg m$^{-3}$). These eBC mass concentrations are close to the average values reported during wintertime for other locations in large urban areas like London (1.3 μg m$^{-3}$; Liu et al., 2014), Las Vegas (1.8 μg m$^{-3}$; Brown et al., 2016) and Fresno in California (1.05 μg m$^{-3}$; Collier et al., 2018), thus representing

typical urban wintertime burdens. The ACSM measured a median organic mass concentration of 1.4 μg m$^{-3}$ (IQR = 2.4 μg m$^{-3}$), higher than any inorganic species (nitrate: 0.8 μg m$^{-3}$, IQR = 1.2 μg m$^{-3}$; ammonium: 0.5 μg m$^{-3}$, IQR = 0.8 μg m$^{-3}$; sulfate: 0.1 μg m$^{-3}$, IQR = 0.1 μg m$^{-3}$ and only traces of chloride). Finally, the wind speed was generally low (median of 0.4 m s$^{-1}$, IQR = 0.9 m s$^{-1}$) with no wind speed higher than 5 m s$^{-1}$, and the temperature varied between -7.8 °C and 14.3 °C, with a median of 4.4 °C (IQR = 5.9 °C).

Figure 4 gives an overview of wind and hygroscopicity parameters as well as mass concentrations of organic and inorganic particulate matter during the period covering the four analyzed fog events. A clear cause of the reduction of the concentration of any type of particles is occurrence of a medium (or high) wind speed, causing a dilution effect (Zhu et al., 2002), e.g. in the morning of 16 December (Fig. 4a-b).

The mobility diameters corresponding to sCCNC-activated fractions of 25%, 50% and 75% derived from sCCNC measurements are plotted in Figure 4c for the three example supersaturations. The dry diameter at 50% sCCNC-activation (referred to as sCCNC-critical diameter $D_{crit}^{sCCNC}$) provides information on the median particle hygroscopicity: for a fixed SS, particles activate at a lower diameter if they are highly hygroscopic, thus



resulting in lower $D_{\text{crit}}^{sCCNC}$, and vice-versa. As $\kappa_{\text{median}}$ is directly calculated from sCCNC-derived $D_{\text{crit}}^{sCCNC}$, the time series of $\kappa_{\text{median}}$ values gives the same type of information but for making results from all nine SS set in the CCNC directly comparable. The particle hygroscopicity at all SS except the highest one (SS=1.33%) was quite low (Fig. 4d), which also applies for the whole campaign with $\kappa_{\text{median}}$ between 0.19 and 0.24 depending on the

SS (Table 1). This indicates a dominant contribution of compounds exhibiting little or no hygroscopic growth such as organics and black carbon possibly emitted by traffic or wood burning. Dominant contribution of non- or only moderately hygroscopic matter (BC, organics) opposed to only minor contribution of hygroscopic inorganic ions is indeed confirmed by the ACSM composition measurements (Fig. S1 and Table 1). Observed aerosol hygroscopicity was in the range of values reported in the literature for field studies in continental sites

influenced by traffic in winter: Paris (France): 0.09 to 0.17 (Hammer et al., 2014b) and 0.08 to 0.24 (Jurányi et al.. 2013); Mexico City (airborne measurements): 0.2 to 0.3 (Shinozuka et al., 2009) and the Pearl River Delta region in China: 0.18 to 0.22 (Jiang et al., 2016).

Mean aerosol hygroscopicity increases with increasing particle size (Table 1), a feature which is often observed for atmospheric aerosols (Swietlicki et al., 2008). Note, the aforementioned trend of $\kappa_{\text{median}}$ with particle size is

broken for the data from measurements at lowest and highest supersaturations; however, this is likely an artefact due to increased uncertainty related to CCN calibration at these two supersaturations.

A closer look at the time series shown in Figure 4d reveals some interesting features. Sometimes, particle hygroscopicity inferred from the measurements at the highest SS drops considerably during the morning rush hour, as will be discussed in more detail in the following section. Exactly the opposite effect, i.e. strongly

increased particle hygroscopicity up to $\kappa_{\text{median}} = 0.6$ at the highest SS (most of the time representative of 25 to 40 nm particles), is often observed between around 1:00 and 10:00 (LT). This increase in $\kappa_{\text{median}}$ could also be seen, though to a lesser extent at SS=0.93%, and at times even down to medium to low SS. The diurnal patterns of $\kappa_{\text{median}}$ averaged over the whole campaign, shown in Figure S2, also reveal increased hygroscopicity in the second half of the night of the smaller particles (high SS), compared to the minimum which occurs in the

afternoon. This shows that these episodes are, while not occurring every night, still relevant for aerosol hygroscopicity on a time-averaged basis. Having said this, the campaign average variability of $\kappa_{\text{median}}$ in terms of IQR is largely independent of SS (particle size), indicating that variations in aerosol composition occurring due to e.g. variations in air mass type or source contributions independent of time-of-day dominate over the systematic but small diurnal pattern.

The cause of the night-time increase of hygroscopicity at smaller particle sizes was not identified, but a probable explanation is the acquisition of ammonium nitrate. An increase in ammonium nitrate volume fraction by condensation would more efficiently proceed for smaller particles, due to their higher surface-to-volume ratio, thereby increasing their hygroscopicity. The fact that the retrieved $\kappa_{\text{median}}$ value reached up to 0.6 at and approached the reported mean $\kappa$ value of ammonium nitrate (0.67; Petters and Kreidenweis, 2007) supports this

hypothesis.

### 3.2 Influence of traffic on aerosol population, mixing state and hygroscopicity

Previous studies enumerated the diversity of aerosol types that are present in European cities (e.g. Putaud et al., 2010), with seasonally variable source contributions to organic carbon (OC) and elemental carbon (EC; Szidat et al., 2006; Gelencsér et al., 2007): although EC mostly originates from fossil fuel combustion in summer,



biomass-burning emissions from residential heating have been reported to represent a significant fraction of EC emissions in winter.

By plotting diurnal cycles of particle number in different diameter ranges and eBC concentrations (Fig. 5a,b), we could identify periods with high concentrations from around 08:00 to 12:00 (LT) during weekdays, peaking from 8:00 to 10:00 (LT). This time window is hereafter referred to as the rush hours, as it generally corresponds to the times when people commute to work during weekdays. We hypothesized that this concentration peak is caused by traffic emissions, rather than the second most common source of BC in Zurich, wood burning (Zotter et al., 2017). To test this hypothesis, we show campaign averaged diurnal patterns of the absorption Ångström exponent (AAE) in Figure 5c. The characteristic values of the AAE for traffic (0.9 to 1) and wood burning (1.47 to 1.80) were previously reported in winter in Zurich (Zotter et al., 2017). In this campaign, the AAE varied between these two ranges, indicating the presence of emissions from both sources. The AAE values are systematically lower during weekdays than weekend days, when almost no heavy duty vehicles, much less light duty vehicles and also less passenger cars are on the road. The minimum AAE value is reached at 10:00 LT. during weekdays, in agreement with the concentration peaks seen in Figure 5a,b. Consistent results are found for the diurnal cycles of the organics to eBC mass ratio shown in Figure 5d. Although both traffic and wood combustion contribute to BC and organics emissions, wood burning emissions are associated with much higher organics to BC ratios (Laborde et al., 2013). The lowest values of organics to eBC mass ratios (close to 1) were found during the rush hours of weekdays when traffic emissions dominate. During night time, when wood burning emissions contributed to a much larger extent, the organics to eBC mass ratio increased to around 1.5.

BC particles freshly emitted from traffic sources are typically less hygroscopic than background aerosols. Therefore, it is expected that the rush hour peak in traffic contribution is also reflected in aerosol mixing state with respect to hygroscopicity, e.g. in data such as these provided by the sCCNC. If all particles sampled were internally mixed, the resulting size-dependent sCCNC-activated fraction would be a step function (slightly inclined because of finite instrumental resolution), with all particles larger than a certain mobility diameter activating and all smaller particles remaining in the interstitial phase (Moore et al., 2010). In contrast, if the sCCNC-activated fraction curve was broadened, i.e. if CCN activation was gradually occurring over a wider range of mobility diameters, this would indicate increased degree of chemical heterogeneity (external mixing). Following the approach of Jurányi et al. (2013), we use the normalized difference between the 25% and 75% activation diameters ($(D_{75}^{sCCNC} - D_{25}^{sCCNC})/ D_{\text{crit}}^{sCCNC}$) at a fixed SS as an indicator of the degree of external mixing state regarding sCCNC-derived particle hygroscopicity in the size range around $D_{\text{crit}}^{sCCNC}$ (see Figure 3b for the retrieval of $D_{75}^{sCCNC}$ and $D_{25}^{sCCNC}$). Figure 4c shows, mostly seen for the highest SS, that the periods with the highest degree of external mixing were the rush hours (around 8:00 to 10:00 LT) of the weekdays, confirming the above statement that freshly emitted traffic emissions are indeed a significant source of small externally mixed and poorly hygroscopic particles. While the non-hygroscopic particles from fresh traffic emissions usually affect $D_{75}^{sCCNC}$ only, even $D_{\text{crit}}^{sCCNC}$ increases for the highest SS in the most extreme cases, e.g. during the rush hours of 15 and 17 December (Fig. 4c). During the rush hours of 16 December 2015, the absence of a clear peak of external mixing can probably be explained by occurrence of high wind speed, which causes efficient dilution of the fresh emissions with background aerosol.

A more comprehensive analysis of the impact of different aerosol source on aerosol mixing state is done by



means of diurnal patterns of the indicator of mixing state variability for four different SS (Fig. 6). The mixing state indicator values at 0.14% SS, corresponding to mobility diameter of approximately 120 to 220 nm, were quite low, exhibited virtually no diurnal variation, and the difference between weekdays and weekend days was almost inexistent. This indicates that the background aerosol consisted for the most part of large, internally

mixed particles. However, with increasing SS, i.e. with decreasing particle mobility diameter, a peak of externally mixed particles resulting in higher mixing state indicator values gradually appeared in the morning rush hours of weekdays. This shows that the diurnal pattern, already seen in Figure 4 for 5 consecutive days, occurs frequently such that it is reflected in the campaign averaged data too. This rush hour peak in degree of external mixing is most pronounced and significant for SS = 0.67% and 1.33%, i.e. in the mobility diameter

range below 100 nm, representing the typical diameter range of traffic-emitted particles (Laborde et al., 2013: Schwarz et al., 2008). The size-dependence of the relative contributions of local and background aerosol was already highlighted by Baltensperger et al. (2002), who measured the particle hygroscopicity in summer in Milan and concluded that particles in the range 50 to 200 nm were mostly externally mixed.

Previous field studies already reported the variability of the mixing state and hygroscopic properties of particles

depending on their source and air mass age: Subramanian et al. (2010) reported a higher degree of external mixing for BC (i.e. thinner coatings) sampled over the city of Mexico than for older background air masses; Cubison et al. (2008) showed that the primary hydrophobic aerosol mass was no longer a significant component of the aerosol mass 1 to 2 days after emission, mainly because of condensation of secondary species).

The influence of traffic and wood burning emissions on sCCNC-activated fraction spectra is further investigated with Figure 7, in which the data set of the whole campaign is temporarily split and separately averaged for high traffic / low wood burning influence (rush hour from 8:00 to 10:00 LT) and low traffic / high wood burning influence (night time from 1:00 to 7:00 LT), according to the diurnal patterns shown in Figure 5. This split is separately done for weekdays and weekend days. The CCN properties of the most hygroscopic fraction of the

aerosol, which is dominated by contributions from the background aerosol and shows up in the range of sCCNC-activated fractions between 0% to around 60% or more, do not significantly differ between high traffic influence and high wood burning influence, nor between weekdays and weekends. By contrast, systematic variations are found for the less hygroscopic aerosol fraction. On weekdays (Fig. 7a), the sCCNC-activated fraction decreased by around 10 to 15% in the mobility diameter range in which the sCCNC-activated fraction is

greater than ~60%, when comparing the traffic dominated periods with the wood burning dominated periods. Furthermore, complete activation is hardly reached during traffic dominated periods, also at the highest SS and largest mobility diameters covered. By contrast, complete activation is reached for particles greater than around 200 nm in mobility diameter during wood burning dominated times. On weekends, the relative contribution of wood burning to BC is higher than at any time on weekdays, based on AAE shown in Figure 5c, due to

substantially less traffic emissions. Consequently, the difference in sCCNC-activated fraction spectra between the rush hour and night time windows largely disappears (Fig. 7b), and all averaged sCCNC-activated fraction spectra become equal to the night time sCCNC-activated fraction spectra during weekdays. These observations show that the fresh BC particles from traffic emissions are very poor CCN, whereas BC-containing particles from wood burning are at least moderately efficient CCN. This is explained by the facts that traffic emits almost

pure BC, whereas BC from wood burning is internally mixed with co-emitted organics. This interpretation is





consistent with the diurnal pattern of the organics to eBC mass ratio shown in Figure 5. It is also consistent with previous urban measurements in Paris, where Laborde et al. (2013) showed the same difference in BC mixing state and hygroscopic growth between these two BC sources, and where Jurányi et al. (2013) showed, using a mixing-state resolved hygroscopicity-CCN closure approach, that the difference in hygroscopic growth results

in a corresponding difference of CCN activity as expected from Köhler theory. Moreover, the largest traffic effect, i.e. decrease of sCCNC-activated fraction, occurred for small particles in the mobility diameter range of 40 to 110 nm corresponding to the size range previously shown to include the majority of BC particles emitted in an urban environment (Schwarz et al., 2008). The traffic effect was much less pronounced at larger mobility diameters (200 to 400 nm), also consistent with findings by Laborde et al. (2013) in Paris.

**3.3 Activation cut-off diameters $D_{50}^{fog}$ and $D_{half}^{fog}$, and effective peak supersaturation $SS_{peak}$ during fog events**

The combination of total and interstitial inlets was used to determine the number fraction of particles that were activated to fog droplets as a function of particle optical diameter. A comparison of the fog-activated fraction spectrum of the bulk aerosol inferred from SMPSs particle number size distributions with the fog-activated

fraction spectrum of BC-free particles (which represent the majority of particles) inferred from SP2 measurements is shown in Figure 8 for the 14 December fog event (and in Figure S5a-c for the other three fog events). The reasonable agreement between the SMPSs-derived and SP2-derived fog-activated fractions suggests that the sizing of these three instruments (SP2 and both SMPSs) is correct and that activation cut-off diameters inferred from SP2 data of BC-free particles are equivalent to those derived from SMPS data (which

was done in previous literature discussed below). Furthermore, the use of the LEO-fit derived results, which is the only option for BC-containing particles, is validated by the good agreement between the fog-activated fractions of BC-free particles derived from the standard scattering signal analysis and from the LEO-fit analysis.

Half and 50% activation cut-off dry diameters ($D_{half}^{fog}$ and $D_{50}^{fog}$; see Sect. 2.3.3 for definitions) were extracted

from the fog-activated fraction spectrum of BC-free particles for each fog event and are shown in Figure 3. The median hygroscopicity parameter inferred from sCCNC measurements was extrapolated to the cut-off diameter range as shown in Figure 3c. Cut-off diameter and corresponding $\kappa_{median}$ are then used as inputs to the $\kappa$-Köhler theory to retrieve the fog $SS_{peak}$ as described in detail in Sect. 2.3.5. Table 2 lists the times and duration of the four fog events analyzed in the present work, as well as measured LWC, number fraction of particles activated

to fog droplets, $D_{half}^{fog}$ and $D_{50}^{fog}$, $\kappa_{median}$ extrapolated to the size range of $D_{half}^{fog}$ and $D_{50}^{fog}$, and $SS_{peak}$. Fog events occurred only when the wind speed was lower than approximately 1 m s$^{-1}$ (Fig. 4a). The impact of wind on fog occurrence was clearly observed in the afternoon of the 14 December fog event: the fog dissipates when the wind speed increased, and a new fog event started when the wind speed decreased again.

Using SMPS measurements behind the total and interstitial inlets, we calculated the fraction of particles

activating to fog droplets and confirmed that they represent a very small subset of the aerosol population. Only for the 18 December fog event the fraction of activated particles in the mobility diameter range between 20 and 593 nm was higher than 1%. This fraction depends on various parameters such as $SS_{peak}$ and particle number size distribution shape; therefore it may vary significantly for other locations and periods.

All four fog events were rather similar in terms of LWC, activation cut-off diameter and $SS_{peak}$, $D_{half}^{fog}$ and $D_{50}^{fog}$





lay in the range 320 to 380 nm and 370 to 470 nm, respectively, which is in very good agreement with the results from Hammer et al. (2014b), who measured a median activation cut-off diameter between 364 and 450 nm during fog events in Paris (they found similar $D_{half}^{fog}$ and $D_{50}^{fog}$ but calculated lower and upper limits with two different methods). The $\kappa_{median}$ of the particles activating to fog droplets were also very close for the first three events (0.16 to 0.18) but lower for the last one (20 December; $\kappa_{median} = 0.12$).

Two values of $SS_{peak}$ are given for each fog event; the lower value was retrieved from $D_{50}^{fog}$, the higher from $D_{half}^{fog}$. $SS_{peak}$ ranged from 0.036% during the 15 December fog event to 0.058% during the event of 18 December. This is in very good agreement with a previous fog study by Hammer et al. (2014b) in Paris during wintertime, who reported $SS_{peak}$ of 0.031 to 0.046% over 16 fog events. The low $SS_{peak}$ in fog are also comparable to the 0.05% $SS_{peak}$ estimated by Schroder et al. (2015) for two low-altitude stratocumulus clouds at the Californian Pacific coast. However, the droplet formation process in these clouds differed in so far as $D_{50}^{cloud}$ was lower (239 and 241 nm) because the $\kappa$-value was higher (0.50 and 0.41; derived from an aerosol mass spectrometer, AMS). Similarly, in low-altitude shallow layer or stratus clouds, Pruppacher et al. (1998) estimated an effective $SS_{peak}$ of approximately 0.05% and Leaitch et al. (1996) reported a maximum threshold value of 0.1%. Modelling results from Ming and Russell (2004), who simulated a fog event, are also very close to ours; they predicted a maximum $SS_{peak}$ of 0.030% and a maximum LWC of 150 mg m$^{-3}$ in the simulation. However, cumulus clouds present much higher $SS_{peak}$ due to the high updraft velocities and variations of pressure during their formation; previous studies by Pruppacher et al. (1998) and Hammer et al. (2014a) reported ranges of 0.25 to 0.7% and 0.37 to 0.5%, respectively. LWC values in convective clouds can also reach much higher values than in fog; Hammer et al. (2014a) and Reid et al. (1999) measured values up to 700 and 2000 mg m$^{-3}$, respectively.

**3.4 Size-dependent activation of BC-containing particles to fog droplets**

Because of the very low $SS_{peak}$ in fog, only large particles activate to droplets. During the four fog events investigated in this study, the minimum particle mobility diameter for activation was roughly 210–300 nm, as shown in Figures 8 and S5a-c. As the largest BC cores were also of about this size (Fig. S4), bare BC cores could anyway not activate to droplets because of being smaller than the activation cut-off for hygroscopic particles.

Figures 8 and S5 show that the droplet activation behaviour of BC-containing particles was very similar to the one of BC-free particles; the presence of BC within the particles did not significantly alter their activation behaviour compared to BC-free particles. This somewhat surprising result can be explained by the fact that the dominant fraction of BC cores is thickly coated, as explained in the Supplement and shown in Figure S3.

The activation of BC containing particles to fog droplets, as a function of BC core mass equivalent diameter, derived from the SP2 data behind total and interstitial inlets is shown in Figure 9 for the 14 December fog event. The same analysis was performed for the 15, 18 and 20 December fog events; results were very similar, they are therefore not shown here. The BC core mass size distribution measured behind the total inlet peaked at a diameter around 140 nm (Fig. 9a). In the range 127 nm $< D_{rBC} < 212$ nm, which overlaps with the peak of the mass size distribution, it is possible to split all BC-containing particles in the two sub-classes of BC particles with "no to moderate coating" or "thick coating" using the delay time method (see 2.2.3). When comparing particles with equal BC core size (Fig. 9b), only around 15 to 25% had thick coating, whereas 75 to 85% had no





or moderate coatings only. This corresponds with expectations for a site with substantial influence from fresh traffic emissions. Within the size limits of the delay time method, the fog-activated fraction of all BC cores shown in Figure 9c was close to zero and, within experimental uncertainty, also identical to the fog-activated fraction of BC cores with thin/moderate coatings because the BC particle population was dominated by this

subclass (Fig. 9b). The BC cores with thin/moderate coating remain, for $D_{rBC} < 212$ nm, smaller than the overall particle diameter above which the fog-activated fraction starts increasing even for BC-free (water-soluble) particles (Fig. 8). By contrast, the fog-activated fraction of thickly coated BC particles gradually increased with BC core diameter and reached around 40% at $D_{rBC} = 210$ nm. This is explained by the fact that the substantial coating increases both size and soluble volume fraction of these BC-containing particles such that some of them

have a size bigger than the fog droplet activation threshold. In the size range $D_{rBC} > 212$ nm, the fog-activated fraction of all BC particles also starts increasing because the threshold coating thickness to cross the activation threshold becomes smaller with increasing core size. These results demonstrate in a qualitative manner how acquisition of coatings makes the BC particles better nuclei for fog droplets by increasing overall particle size and solubility.

Besides activation of BC containing particles to droplets, also coagulation between BC-containing particles and existing fog droplets could potentially explain the presence of BC in fog droplets: The probability of coagulation between two particles is increased when the difference between their respective diameters increases, so small BC-containing particles may potentially coagulate with fog droplets. However, Figure 9c clearly shows that droplet activation of BC-containing particles is the mechanism that explains the incorporation of BC cores into

fog droplets in the present study: if coagulation between BC particles and fog droplets was giving a dominant contribution, then the fog-activated fraction of BC particles would exhibit much less size and coating dependence and rather with opposite trends.

### 3.5 Linking mixing state of BC, fog-critical supersaturation and droplet activation

The fog-critical supersaturation of individual BC-containing particles was calculated using $\kappa$-Köhler theory and

particle properties constrained with SP2 and sCCNC data: the former providing particle size and BC volume fraction, the latter providing coating hygroscopicity (see Sect. 2.3.2; Eq. (1) and (3)). Figure 10a shows these $SS_{crit}$ arranged by BC core size (abscissa) of all individual BC-containing particles as inferred from the 14 December fog event data as an example: data points from BC particles sampled behind the interstitial inlet are colored by coating thickness, those sampled behind the total inlet are shown as grey dots. The fog effective peak

supersaturation ($SS_{peak}$) retrieved from $D_{50}^{fog}$ using the method described in Sect. 2.3.5, is marked with a blue horizontal line. Theoretically, every BC-containing particle whose $SS_{crit}$ is below the fog $SS_{peak}$ should activate to a fog droplet (i.e. no data point from the interstitial inlet should appear below the blue line), whereas those particles with $SS_{crit}$ greater than the fog $SS_{peak}$ should remain interstitial (i.e. equal number of data points for interstitial and total inlets above the blue line). The ratio of interstitial to total BC particle number indeed

decreases below the blue line. However, some BC particles with $SS_{crit} < SS_{peak}$ are still detected behind the interstitial inlet, which can be explained by several facts: first, the fog $SS_{peak}$ is not perfectly constant during the event; second, the input parameters for calculating $SS_{crit}$ are tainted with random measurement noise on single particle level; third, potential shortcomings of the $\kappa$-Köhler theory such as neglecting variations in particle size.



In order to achieve a more quantitative closure between expected and observed activation of BC-containing particles to fog droplets, the single particle data as shown in Figure 10a for one example fog event, were aggregated into SS bins ($\Delta SS_{crit} = 0.01\%$) and averaged over all core sizes. The same was done for each fog event and resulting activation curves are shown in Figure 10b-e.

For each fog event, 50% fog-activated fraction is reached at an $SS_{crit}$ very close to the $SS_{peak}$ derived from $D_{50}^{fog}$. This agreement confirms that observed activation of BC particles in the fog matches the expected droplet activation behaviour of BC-containing particles as theoretically predicted from independently measured BC-particle properties (size, BC volume fraction and coating hygroscopicity). This demonstrates that closure is successfully achieved, i.e. SP2-based characterization of BC-containing particle properties combined with $\kappa$-

Köhler theory is sufficient to accurately describe the activation behaviour of BC-containing particles in fog, despite the fact that either of them are based on the simplifying assumption of spherical core-shell morphology.

Figure 10b-e also contains the fog-activated fraction of BC-free particles detected by the SP2, for which $SS_{crit}$ was calculated using $\kappa$-Köhler theory with $\kappa_{median}$ and optical diameter from the SP2. 50% activation is by

definition reached by those particles with $SS_{crit}$ equal to $SS_{peak}$ inferred from $D_{50}^{fog}$ (small deviations are explained by binning the fog-activated fraction data in supersaturation rather than diameter space). The fact that the activation curves of BC-containing particles in Figure 10b-e agree well with the activation curves of BC-free particles implies the following: the activation of BC-containing particles to fog droplets can be described identical to the activation of BC-free particles but for adjusting the $\kappa$-value with the ZSR-rule to account for the

volume fraction of insoluble BC. This is an alternative but equivalent view of how closure is achieved for the activation of BC to fog droplets.

To our knowledge, Schroder et al. (2015) performed the only other similar closure between critical supersaturation of atmospheric BC-containing particles and retrieved peak supersaturation of real clouds, low-

altitude stratocumulus in their case. They selectively sampled cloud droplets using a counterflow virtual impactor (CVI) and a total inlet and calculated particles $SS_{crit}$ an cloud $SS_{peak}$ in a very similar way as in the present study (extracting $D_{rBC}$ and $\Delta_{coating}$ from SP2 measurements and $\kappa_{median}$ from AMS measurements; and using the ZSR mixing rule together with the $\kappa$-Köhler theory). However, while reporting general agreement between the range of predicted particle $SS_{crit}$ and calculated cloud $SS_{peak,}$ they could not achieve an unequivocal

closure but found as they could only determine lower limit values for $\Delta_{coating}$ because of technical issues of the SP2.

The activation curves shown in Figure 10b-e are, despite being based on single particle data, averages over all BC-containing particles with equal $SS_{crit}$, i.e. over particles with different combinations of BC core and coating

thickness. In order to assess the role of BC core size and coating thickness for fog droplet activation in more detail, the fog-activated fraction inferred from measurements behind interstitial and total inlets, is shown in Figure 11 as a function of BC core mass equivalent diameter and coating thickness measured by the SP2. This figure quantifies the coating thickness that was necessary for a BC core of a certain size to make it activate to a fog droplet at the $SS_{peak}$ prevalent in the fog event under investigation. For a fixed BC core diameter, the fog-

activated fraction generally increases from zero (blue color) to 100% (red colour) for increasing coating





thickness, i.e. along vertical lines across the images. As expected and already qualitatively shown in Figure 9, BC cores with thin to moderate coatings remained interstitial, with the threshold coating thickness for 20% fog-activated fraction (blue color) increasing from $\Delta_{coating} \approx 20$ to $50$ nm to $\Delta_{coating} \approx 80$ to $120$ nm for BC core diameters of 300 nm and 60 nm, respectively. Equivalently, the threshold coating thickness required for 100%

fog-activated fraction (red color) also increases with decreasing BC core diameter, as particle size favors droplet activation (McFiggans et al., 2006). The dashed lines in Figure 11 represent pairs of $D_{rBC}$ and $\Delta_{coating}$ that would theoretically give to a particle an $SS_{crit}$ equal to the minimal (upper line) and maximal (lower line) estimations of $SS_{peak}$ during the related fog event. This theoretical activation limit differs between fog events due to variations in $SS_{peak}$ and $\kappa_{coating}$. Generally, the agreement between the theoretically expected threshold and the observed

50% fog-activated fraction (green color) is good across all BC core diameters and fog events, which means that closure is achieved for all BC core sizes covered by the measurement. Systematic deviations may possibly be present for the largest core sizes, where the activation threshold is at highest BC volume fractions and where considering interactions between the BC core and the first layers of water surrounding it might become important. However, these differences are minor, counting statistics in this range is too poor, and simplifying

assumptions in the data processing are too numerous to justify use of more sophisticated theoretical calculations. The overall good agreement validates the simplified theoretical description as explained in Sect. 2.3.2 and confirms that, within uncertainties, the fate of a BC particle in fog can be predicted if $D_{rBC}$, $\Delta_{coating}$ and $SS_{peak}$ are known.

Dalirian et al. (2018) showed in a laboratory study that the size and mixing state dependence of CCN activation of artificially produced BC with variable coatings agrees well with theoretical predictions. Our study shows that their findings from the laboratory also hold for the behaviour of atmospheric BC during fog formation, thus justifying application of theories based on Köhler theory and ZSR mixing rule, or of parametrizations of e.g. activation thresholds for BC particles derived from such theories, in atmospheric model simulations.

**4 Conclusions**

During winter 2015/16 a field campaign was performed at the Irchel University campus in Zurich in order to investigate the size-dependent activation of BC at different mixing states. We distinguished fresh BC-containing particles from emissions in the vicinity of the measurement site (heavily used roads, residential heating) and aged, background BC-containing particles, and found, based on the sCCNC-activated fraction spectrum,

evidence of high degree of external mixing during the morning rush hours due to a substantial number fraction of non-hygroscopic particles from fresh traffic emissions.

The half-activation cut-off diameter for activation of BC-free particles to form fog droplets varied between 320 and 380 nm during four fog events, which translates to very small overall activated number fractions in the range of ~1%. Fog peak supersaturations, which were inferred by combining this activation cut-off with CCNC

derived particle hygroscopicity, were found to be between 0.036 and 0.047%, consistent with previous literature. The activation of BC-containing particles to fog droplets was also quantified. The majority of the BC-containing particles remained interstitial (as with the large majority of non-BC-containing particles), only those with substantial coatings were activated as the coating decreases their critical supersaturation by increasing size and



solubility. The threshold coating thickness required for activation was shown to decrease with BC core size, as expected. Quantitative closure between measured and predicted activation of BC-containing particles to fog droplets was successfully achieved. Predictions are based on Köhler theory combined with the ZSR mixing rule constrained with independently measured BC particle properties on single particle level. This confirms that the

activation behaviour of atmospheric BC can be accurately predicted, within experimental uncertainty, with the knowledge of BC core diameter, coating thickness and coating hygroscopicity, while neglecting particle shape effects. When coupling such simplified theoretical descriptions with aging schemes in particle-resolved models, these results may help to reach a more realistic representation of the life cycle of BC in the atmosphere and to narrow the uncertainties associated with estimates of its radiative forcing.

*Acknowledgements*

We would like to thank the University of Zurich for providing access to the Irchel campus as a measurement site and the group led by Ulrike Lohmann ETH Zurich for lending us a CCNC. This work is supported by the ERC under grant ERC-CoG-615922-BLACARAT and the EU FP7 project BACCHUS (grant no. 603445).

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





**Table 1: Statistics of various measured and inferred parameters from data covering the whole campaign.**

| | Unit | Mean | Median | 25th percentile | 75th percentile |
|---|---|---|---|---|---|
| **Total particle number concentration ($D$>7 nm)** | [# cm$^{-3}$] | 6324.2 | 5879.1 | 3584.7 | 9551.5 |
| **CCN number concentration for SS=0.14%** | [# cm$^{-3}$] | 1070.6 | 989.0 | 557.2 | 1449.7 |
| **CCN number concentration for SS=0.21%** | [# cm$^{-3}$] | 1812.0 | 1645.8 | 929.8 | 2524.7 |
| **CCN number concentration for SS=0.27%** | [# cm$^{-3}$] | 2284.9 | 2129.3 | 1181.4 | 3188.6 |
| **CCN number concentration for SS=0.34%** | [# cm$^{-3}$] | 2604.2 | 2405.2 | 1332.6 | 3629.9 |
| **CCN number concentration for SS=0.40%** | [# cm$^{-3}$] | 2892.1 | 2636.6 | 1472.1 | 4042.0 |
| **CCN number concentration for SS=0.47%** | [# cm$^{-3}$] | 3139.4 | 2856.6 | 1671.6 | 4363.9 |
| **CCN number concentration for SS=0.67%** | [# cm$^{-3}$] | 3813.3 | 3392.8 | 2053.2 | 5256.5 |
| **CCN number concentration for SS=0.93%** | [# cm$^{-3}$] | 4403.1 | 3867.4 | 2369.5 | 6228.6 |
| **CCN number concentration for SS=1.33%** | [# cm$^{-3}$] | 5418.4 | 4603.0 | 2865.8 | 7396.9 |
| **eBC mass concentration** | [μg m$^{-3}$] | 1.3 | 1.1 | 0.5 | 1.8 |
| **Organics mass concentration** | [μg m$^{-3}$] | 1.7 | 1.4 | 0.6 | 3.1 |
| **$NH_4^+$ mass concentration** | [μg m$^{-3}$] | 0.5 | 0.5 | 0.2 | 1 |
| **$NO_3^-$ mass concentration** | [μg m$^{-3}$] | 0.9 | 0.8 | 0.3 | 1.5 |
| **$SO_4^{2-}$ mass concentration** | [μg m$^{-3}$] | <0.1 | <0.1 | 0.0 | 0.1 |
| **Hygroscopicity parameter $\kappa_{median}$ for SS=0.14%** | [-] | 0.23 | 0.21 | 0.16 | 0.29 |
| **Hygroscopicity parameter $\kappa_{median}$ for SS=0.21%** | [-] | 0.27 | 0.24 | 0.18 | 0.33 |
| **Hygroscopicity parameter $\kappa_{median}$ for SS=0.27%** | [-] | 0.26 | 0.24 | 0.17 | 0.33 |
| **Hygroscopicity parameter $\kappa_{median}$ for SS=0.34%** | [-] | 0.24 | 0.22 | 0.17 | 0.30 |
| **Hygroscopicity parameter $\kappa_{median}$ for SS=0.40%** | [-] | 0.23 | 0.21 | 0.16 | 0.28 |
| **Hygroscopicity parameter $\kappa_{median}$ for SS=0.47%** | [-] | 0.21 | 0.20 | 0.15 | 0.26 |
| **Hygroscopicity parameter $\kappa_{median}$ for SS=0.67%** | [-] | 0.21 | 0.19 | 0.14 | 0.25 |
| **Hygroscopicity parameter $\kappa_{median}$ for SS=0.93%** | [-] | 0.21 | 0.19 | 0.14 | 0.26 |
| **Hygroscopicity parameter $\kappa_{median}$ for SS=1.33%** | [-] | 0.25 | 0.22 | 0.17 | 0.30 |
| **Temperature 3 m above ground** | °C | 3.6 | 4.4 | 1.2 | 7.2 |
| **Wind speed 3 m above ground** | [m s$^{-1}$] | 0.6 | 0.4 | 0 | 0.9 |



**Table 2: Details of the four analyzed fog events. Uncertainties in brackets are provided as relative errors. Uncertainties of the input parameters ($D_{half}^{fog}$ or $D_{50}^{fog}$ and $\kappa_{median}$) were propagated using the Monte Carlo method to obtain uncertainties for $SS_{peak}$. The temperature was not varied in these simulations as it has a second order influence on droplet activation compared to $D_{half}^{fog}$ or $D_{50}^{fog}$ and $\kappa_{median}$.**

| | Unit | Type of uncertainty | 14 Dec. | 15 Dec. | 18 Dec. | 20 Dec. |
|---|---|---|---|---|---|---|
| **Start date and time (LT)** | / | / | 14 Dec. 04:30 | 14 Dec. 17:20 | 18 Dec. 01:30 | 19 Dec. 21:50 |
| **End date and time (LT)** | / | / | 14 Dec. 12:00 | 15 Dec. 06:20 | 18 Dec. 05:00 | 20 Dec. 11:40 |
| **Duration** | [h] | / | 7.5 | 9 | 3.5 | 14 |
| **Median liquid water content (LWC)** | [μg m$^{-3}$] | From Allan et al. (2008) | 107 (±20%) | 116 (20%) | 133 (±20%) | 136 (±20%) |
| **Number fraction of particles activated to fog droplets in the $D_{dry}$ range 20 to 593 nm** | [%] | Based on out-of-cloud measurements | 0.6 (±12%) | 0.8 (±7%) | 1.1 (±11) | 0.5 (±14%) |
| **Half/50% activation cut-off diameter in fog ($D_{half}^{fog}$ / $D_{50}^{fog}$)** | [nm] | Based on out-of-cloud measurements | 370-430 (±18%) | 380-450 (±17%) | 320-370 (±20%) | 380-470 (±18) |
| **Hygroscopicity parameter $\kappa_{median}$ extrapolated to the activation cut-off diameters ($\kappa_{median}(D_{half}^{fog})$ and $\kappa_{median}(D_{50}^{fog})$)** | [-] | 95% confidence on extrapolated fit (see Figure 3c) | 0.16 (±56%) | 0.18 (±95%) | 0.17 (±47) | 0.12 (±108%) |
| **Effective peak supersaturation ($SS_{peak}$) in fog retrieved from $D_{50}^{fog}$ and $D_{half}^{fog}$, respectively** | [%] | Monte Carlo method | 0.040-0.051 (±66%) | 0.036-0.046 (±82%) | 0.047-0.058 (±64%) | 0.040-0.055 (±83±) |



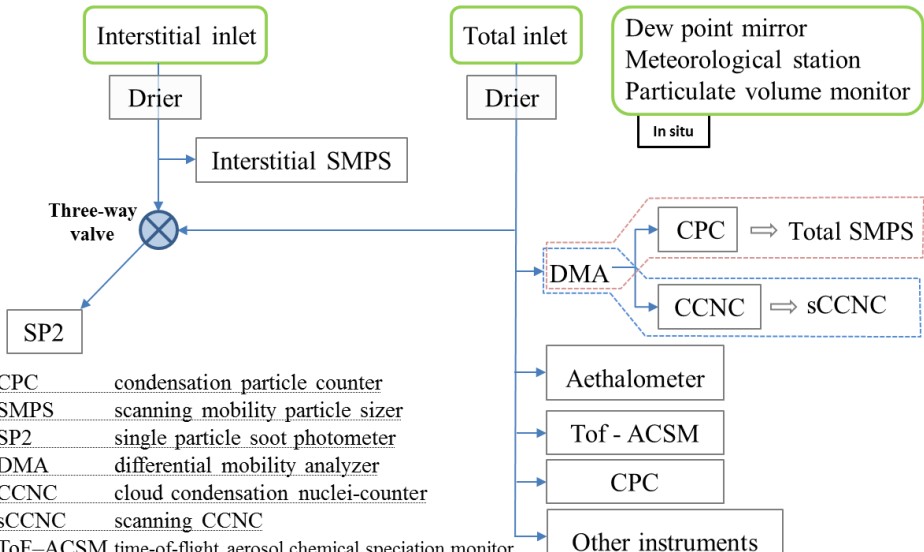

**Figure 1: Scheme of the instrumental setup. The SMPS is a DMA-CPC assembly and the sCCNC a DMA-CCNC assembly.**



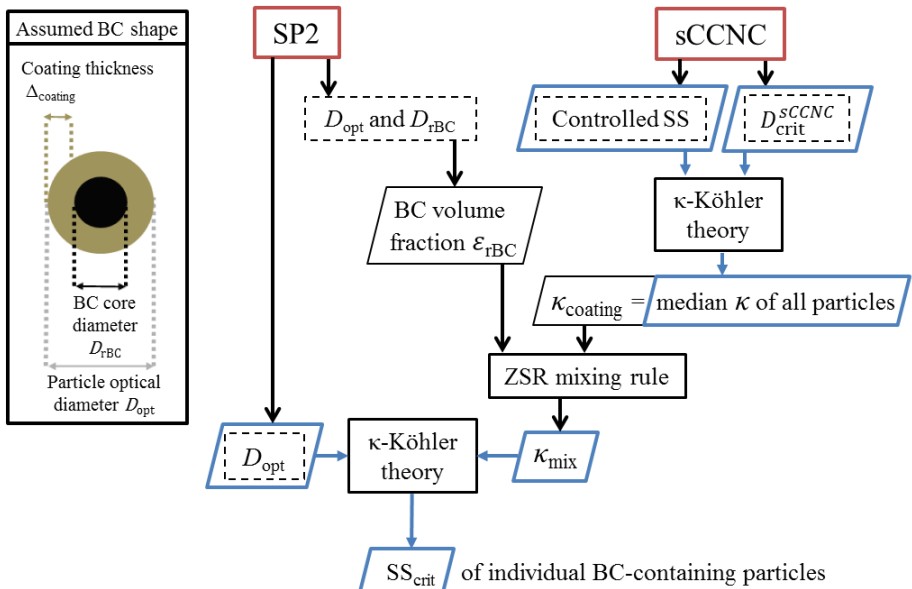

**Figure 2: Approach used to retrieve the SS$_{crit}$ of individual BC-containing particles. Red rectangles show the instruments providing the basic input parameters shown in dashed black rectangles. Blue parallelograms depict the input and output parameters of the κ-Köhler theory. Values of κ$_{coating}$ for individual particles are assumed to be equal to the ensemble median κ of all particles at a given size (κ$_{median}$) as derived from the sCCNC and the total SMPS data (see Sect. 2.3.4). Coated sphere morphology is assumed for both interpreting SP2 scattering signals and in κ-Köhler theory.**





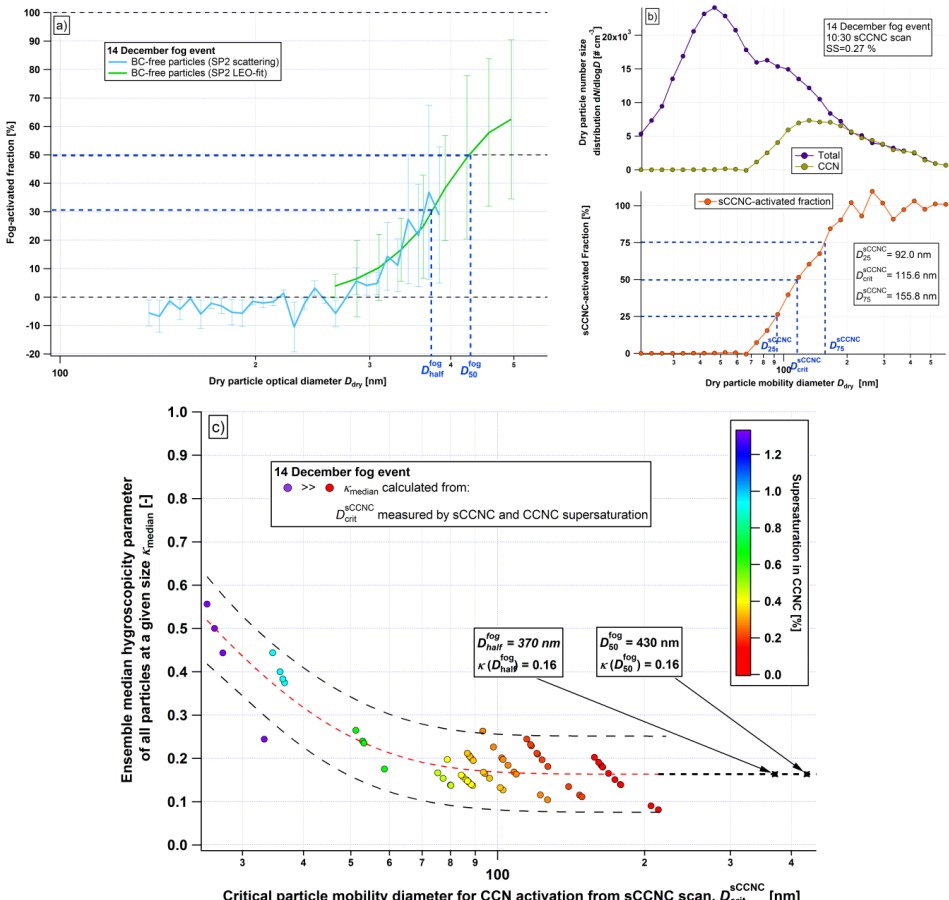

**Figure 3: Example data from the 14 December fog event. (a) sCCNC-activated fraction spectrum including $D_{50}^{fog}$ and $D_{half}^{fog}$ from SP2 measurements. $D_{50}^{fog}$ and $D_{half}^{fog}$ along with $\kappa_{median}$ values from sCCNC measurements are used to obtain two estimates of fog $SS_{peak}$. (b) Total particle and CCN number size distribution from sCCNC measurement and corresponding sCCNC-activated fraction spectrum at a fixed SS used to infer $D_{crit}^{sCCNC}$ and the corresponding $\kappa_{median}$ value. The normalized difference between the 25 % and 75 % activation cut-off diameters, ($D_{75}^{sCCNC}$-$D_{25}^{sCCNC}$)/ $D_{crit}^{sCCNC}$), is used as an indicator of aerosol mixing state. (c) $\kappa_{median}$ values calculated from individual $D_{crit}^{sCCNC}$ retrieved from the sCCNC scans plotted against $D_{crit}^{sCCNC}$ on the abscissa . The points are coloured by the SS applied in the sCCNC. The red line indicates an exponential fit surrounded by 95 % confidence intervals, while the black dashed line indicates an extrapolation of $\kappa_{median}$ to $D_{crit}^{sCCNC}$ corresponding to $D_{half}^{fog}$ and $D_{50}^{fog}$. Note: the peculiar size dependence of the $\kappa$-value, which exhibits increasing hygroscopicity with decreasing particle size, has been observed during the fog events and also few other fog-free nights covered in this study (see Figure 4). However, this feature at the small size end is not relevant for fog droplet activation nor is it representative of the campaign averaged data (see Table 1).**





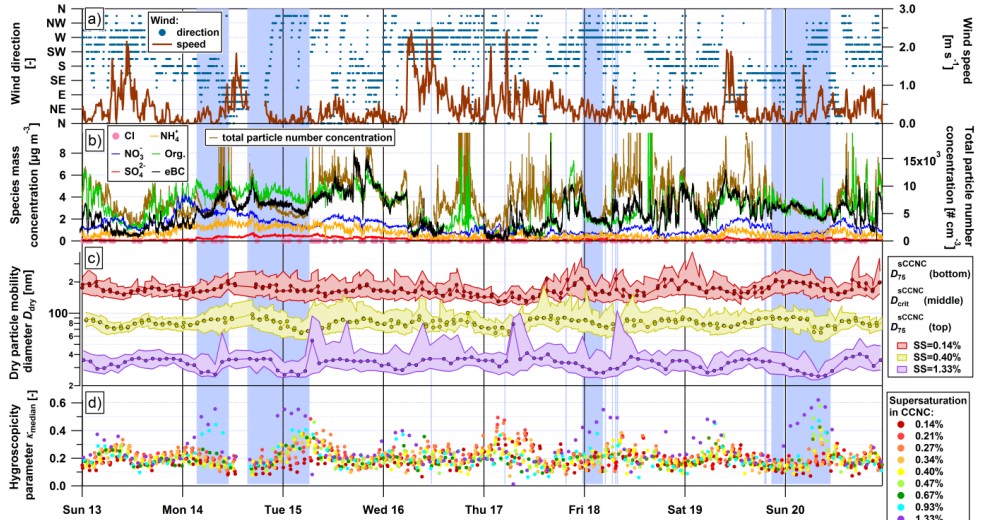

**Figure 4: Time series of various parameters during the period of the analyzed fog events (blue shadings) including (a) wind speed and direction; (b) mass concentrations of organics, inorganic species, eBC as well as total particle number concentration ($D$ > 7 nm) (c) Dry particle mobility diameters corresponding to 25 %, 50 % ($D_{\mathrm{crit}}^{sCCNC}$) and 75 % sCCNC-activated fraction at the SS applied in the sCCNC and (d) retrieved $\kappa_{\mathrm{median}}$ value for each sCCNC scan. Note, the measurements at different SS are representative of different particle sizes (see panel (c)). Thus, the dependence of the $\kappa_{\mathrm{median}}$ values shown in panel (d) primarily reflects the size dependence of particle hygroscopicity.**





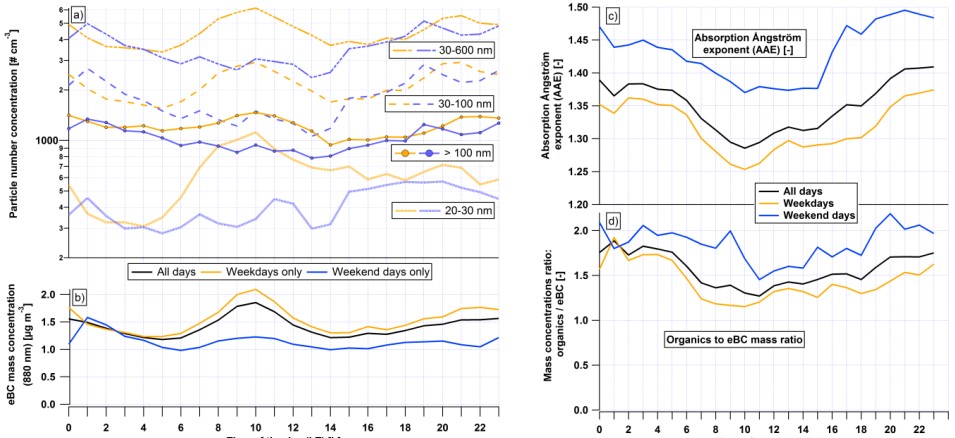

**Figure 5: Diurnal patterns for the whole campaign of (a) Number concentration of particles in the nucleation mode (20 to 30 nm), Aitken mode (30 to 100 nm), accumulation mode (>100 nm) and all particles inferred from integrated SMPS data, (b) eBC mass concentration inferred from the aethalometer measurement at 880 nm, (c) absorption Ångström exponent (AAE) calculated from aethalometer measurements at 470 and 880 nm, and (d) organics (from ACSM) to eBC mass concentration ratio. Substantial differences between values of diurnal cycles at 1:00 and at 2:00 can be seen mainly for weekend days, and to a minor extent for weekdays. They are caused by the discontinuities at Friday and Sunday midnight (later corrected to 1:00 from UTC to LT) and limited statistics (particularly for weekend days).This is also the case for Figure 6.**





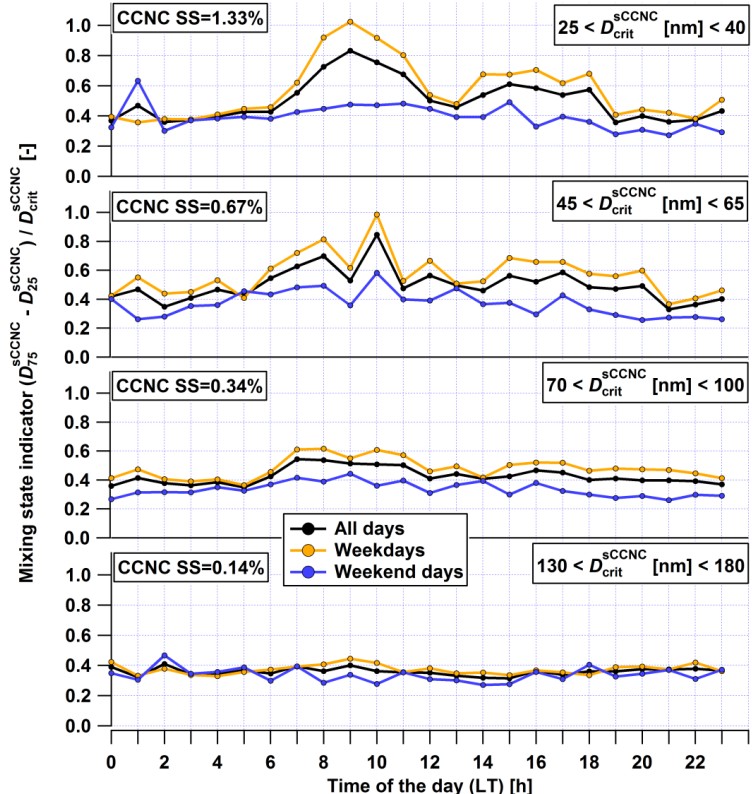

**Figure 6: Diurnal patterns of ($D_{75}^{sCCNC}$ - $D_{25}^{sCCNC}$)/ $D_{crit}^{sCCNC}$) from sCCNC measurements as an indicator of the particle mixing state, averaged during the whole campaign (the larger the value the more externally mixed with respect to hygroscopicity). Results at four different SS are separately averaged over the whole campaign including all days, weekdays only or weekend days only. The approximate ranges of $D_{crit}^{sCCNC}$ corresponding to the respective SS are indicated on each panel.**



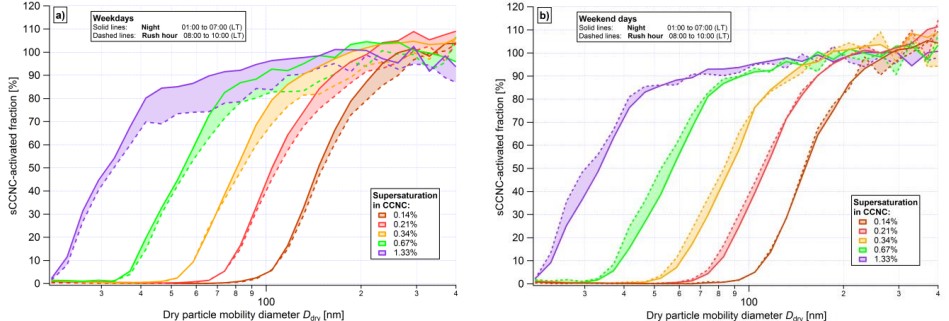

**Figure 7: Averaged sCCNC-activated fraction spectra (from sCCNC measurements). The data set of the whole campaign is temporarily split by (a) weekdays versus (b) weekend days, and also by nighttime versus morning rush hour.**



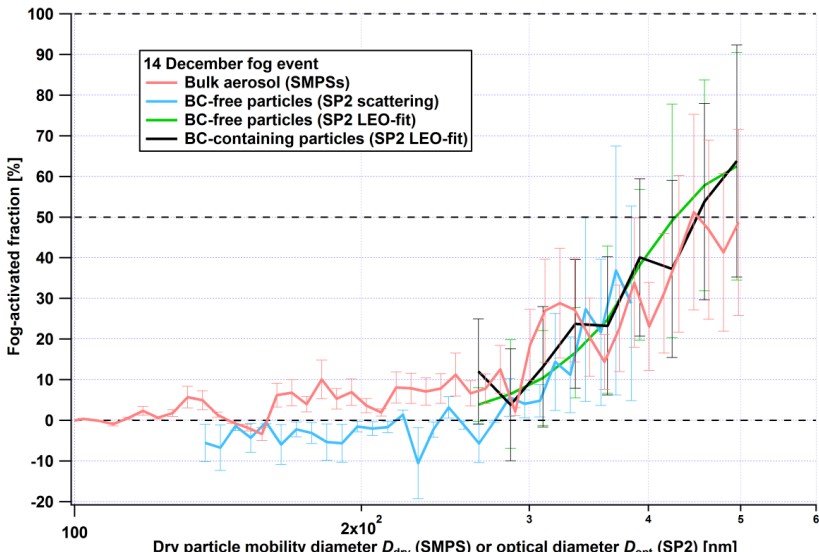

**Figure 8: Fog-activated fraction of the bulk aerosol (from total and interstitial SMPS, red line), BC-containing particles (using SP2 LEO-fit, black line) and BC-free particles (using SP2 scattering signal: light blue line, and SP2 LEO-fit: turquoise line) as a function of the dry particle mobility diameter (for SMPS data) and optical diameter (for SP2 data) during the 14 December fog event. The 1-σ uncertainties of the BC-containing particle data are Poisson-based with respect to the BC core number size distribution; the other ones are dominated by the level of (dis-)agreement of the interstitial and total measurements, which was determined during out-of-cloud periods and propagated through the calculation of fog-activated fraction.**





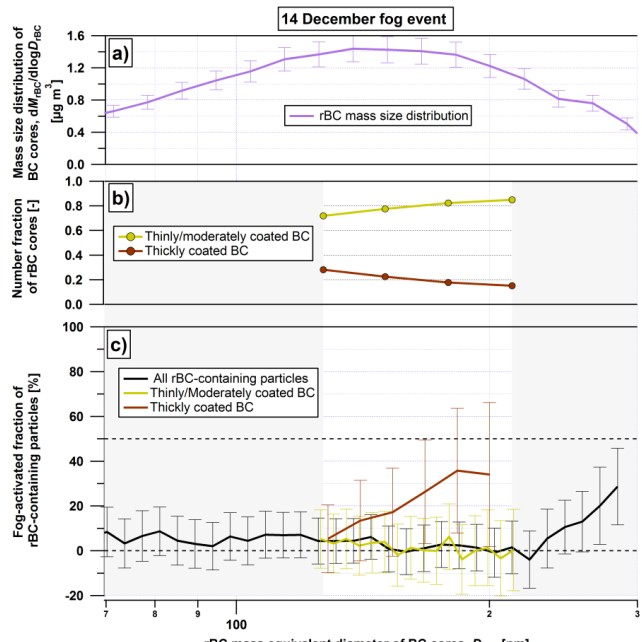

**Figure 9: BC particle properties and behaviour during the December 14 fog event as a function of rBC mass equivalent diameter of the BC cores. (a) rBC mass size distribution (total inlet), (b) number fractions of BC-containing particles split in two mixing state classes (total inlet) and (c) fraction of BC-containing particles activated to fog droplets (based on alternating measurements behind the interstitial and total inlets). The two mixing state classes shown in panels (b) and (c) are distinguished by either thick or thin-to-moderate coatings based on the delay time method applied to the SP2 raw data. The 1-σ uncertainties in panel (c) are Poisson-based counting statistics for the rBC core number size distributions propagated through the equation for the fog-activated fraction. The mixing-state resolved data are only shown for the mass equivalent diameter range 127 nm < $D_{rBC}$ < 212 nm, in which detection limits of the SP2 do not introduce any systematic bias. The most thickly coated particles which caused saturation of the scattering signal were included in the subset of BC particles with thick coatings.**



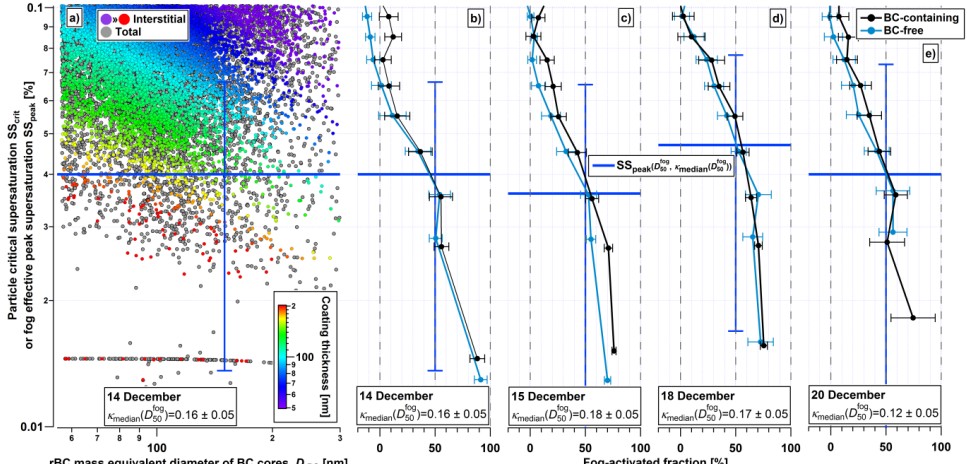

**Figure 10: (a):** $SS_{crit}$ of individual particles sampled behind the total inlet (grey dots) and interstitial inlet (dots coloured by $\Delta_{coating}$) as a function of their $D_{rBC}$ during the 14 December fog event. The distinct band of data points appearing with an $SS_{crit}$ of 0.015 % corresponds to BC-containing particles which caused saturation of the scattering detector even in the leading edge range of the signal, making it impossible to accurately determine $SS_{crit}$. As these particles are known to have lower $SS_{crit}$ than the most thickly coated particles which did not cause signal saturation, they are assigned a "randomly chosen" low value for $SS_{crit}$ and included in the figure. **(b), (c), (d), (e):** fog-activated fractions of BC-containing (black lines) and BC-free (light blue lines) particles per class of 0.01 % SS for the 14, 15, 18 and 20 December fog events, respectively. The variability in the fog-activated fraction induced by the choice of $\kappa_{coating}$ (retrieved $\kappa_{median}\pm0.05$) is represented by horizontal bars. The values of $SS_{peak}$ retrieved using $D_{50}^{fog}$ (with the method explained in Sect. 2.3.5) are marked by horizontal blue lines for each fog event. The lowest part of uncertainty bars is hidden for the 15 and 18 December fog events; they reach 0.0065 % and 0.0068 %, respectively.





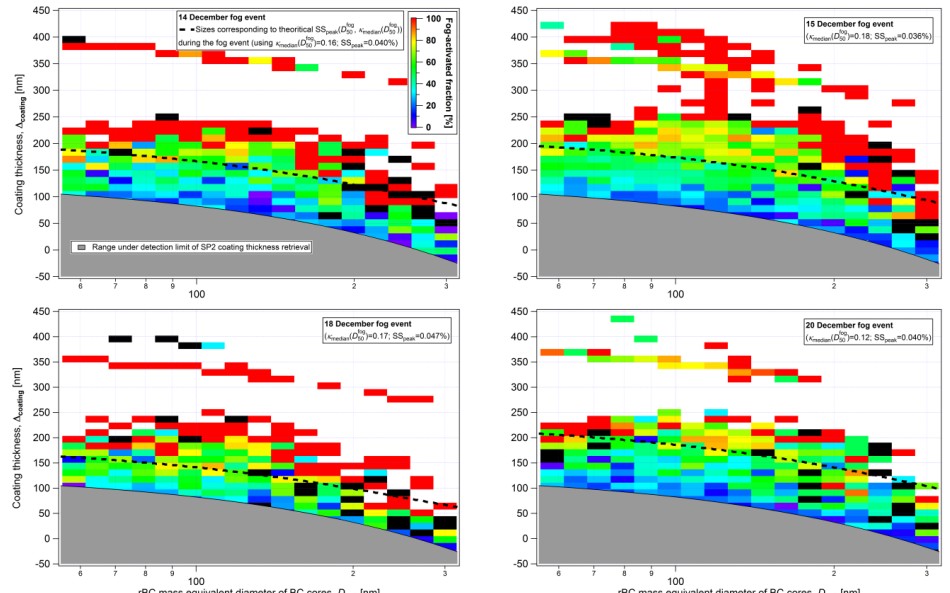

**Figure 11: Fog-activated fraction of BC cores (colour scale), i.e. number fraction of BC particles that formed a fog droplet according to the measurements behind the interstitial and total inlets, as a function of BC core mass equivalent diameter ($D_{rBC}$) and of coating thickness ($\Delta_{coating}$), separately shown for all four fog events. The dashed line shows where the fog-activated fraction is expected to be 50 % according to predictions based on ZSR-rule and $\kappa$-Köhler theory. Black pixels in the image indicate 2D-bins for which no particle was found in the total inlet data while at least one particle appeared in the interstitial inlet data, thus leading to a negative fog-activated fraction. The grey shadings mask the range that is below the detection limit of the SP2, i.e. $D_{opt}$ below around 210 nm. The distinct band of data points at $\Delta_{coating} > \sim 300$ nm appears for the same reason as the band at $SS_{crit} = 0.015$ % in Figure 10a; see corresponding caption for explanation.**



**Authors contribution**

G.M., J.S. were the main organizers of the field campaign; J.C.C. and M.Z. performed instrumental calibration and tuning. All co-authors took part in the data analysis and discussions of the results. G.M. prepared the manuscript with contribution from all co-authors.

The authors declare that they have no conflict of interest.