# Peer review of "Droplet activation behaviour of atmospheric black carbon particles in fog as a function of their size and mixing state"

_Atmospheric Chemistry and Physics, 2018_

## Referee Comment (RC1) · Anonymous Referee #3 · 23 Oct 2018

General comments: This paper investigates the activation of internally mixed black carbon in fog by making use of the low supersaturations within fog to do a closure study on the droplet activation behavior of BC-containing particles. The measurements were taken during a field campaign in a residential area of Zurich in the winter, and indicate that aerosols sourced from traffic during rush hour periods are generally less hygroscopic than aerosols sourced from wood burning.

The paper is well-written and uses novel methods to demonstrate good agreement between predicated and observed behavior. It is appropriate for ACP and is a useful scientific result that will help to constrain the lifetime of BC in the atmosphere, and

demonstrates that simple parameterizations of hygroscopicity in terms of a kappa-Köhler parameter are in good agreement with atmospheric observations.

The methods and measurements are adequately described, as are comparisons with previous atmospheric observations. There are a few minor points that should be clarified to make the paper clearer. The paper would also benefit from a more focused discussion on the major conclusions of the paper, as it is sometimes challenging to follow.

Specific comments: Some of the figures are hard to read (the text is very small). There are also quite a large number of figures (11) and I would suggest moving some of the less important figures (e.g. figures 4, 5, or 6) to the supplemental information to draw more attention to the other figures.

To improve the clarity of the discussion it would be useful to have a table summarizing the different variables, such as the activation diameters and supersaturations.

It would be useful to clearly state the upper and lower limits for the optical size range of non-BC containing particles detected by the SP2 in the 8-channel configuration, and at what optical size the scattering detectors are saturated.

It looks like the laser power in the SP2 used to determine the optical size was only calibrated twice with PSL's, before and after the campaign; were these two calibrations consistent?

Why was the AMS not used to estimate the index of refraction of the coatings based on the chemical composition of the bulk aerosols? Also, what is the motivation behind choosing the refractive index values for the coatings? These values were given without justification or reference. How much would the index of refraction vary based on the observed bulk aerosol chemical composition, and what is the sensitivity of the calculated kappa values for different values of index of refraction for the BC coating?

Figure 9 – This size dependence could also potentially be explained by dry deposi-
tion removing larger, thickly coated BC particles more efficiently. It would be useful to estimate the relative importance of dry deposition. Also, are there any potential size-dependent biases in using the delay time SP2 method for separating the two populations of aerosols?

---

## Referee Comment (RC2) · Anonymous Referee #2 · 29 Oct 2018

**General comments:**

The authors report results from a case study comprising four separate fog events observed in an urban environment in Zurich. Overall, the manuscript is well written and the data analysis has been conducted with great care. The results show that soluble coating on top of an insoluble black carbon (BC) cores indeed increases their ability to serve as condensation nuclei for fog droplets, and the threshold coating thickness decreases with increasing BC core size. Furthermore, the authors demonstrate that a simple $\kappa$-Köhler model can be used to predict the fog droplet activation when the particle size, coating thickness and hygroscopicity of the coating material are known.

[Figure]

Understanding the mixing state of ambient BC and its impact and fate in the atmosphere has been of great interest to aerosol community, and thus, the manuscript by Motos et al. is well within the scope of ACP. That said, the main findings of this study are more incremental rather than novel and (as such) provide a little new insight into the studied topic. Therefore, I would like to see more discussion concentrating on the implications of the results, e.g., how black carbon and its aging are currently treated in particle-resolved models (that were also mentioned in the conclusions) and how these new results could possibly improve these aspects. In other words, there is definitely no need to shift the focus of the paper from experimental research into modelling, but instead, highlight the importance of the results and point out more concretely how aerosol community could benefit from them. In my opinion, this would improve the impact of the paper substantially. Otherwise, I only have a few minor comments and suggestions to be considered by the authors.

**Specific comments:**

**Page 3, Line 21:** A relatively recent paper by Maalick et al. (2016) presents results from LEM simulations concentrating on the effect of BC on the evolution and lifetime of radiation fog. Although this specific paper does not directly deal with BC mixing state, it points out an important aspect of BC in aerosol-cloud/fog interactions and could be cited in this paragraph (if the authors wish).

**Page 3, Line 35:** The study by Dalirian et al. (2018) has been conducted by atomizing BC particles from aqueous solutions and then coating them with organics by using a tube furnace. Therefore, it should be referred to as laboratory study rather than a conventional chamber measurement.

**Page 5, Line 26:** Later in the paper, the authors are referring to uncertainties in CCN calibration (Sect. 3.1). Therefore, it would be good to briefly describe how the instrument was actually calibrated and how the possible instrumental limitations are affecting the measurement uncertainties especially at the lowest and highest supersaturations.

**Page 8, Line 24:** Here, the authors define that the hygroscopicity of the soluble coating $\kappa_{coating}$ is equal to $\kappa_{median}$, which according to Sect. 2.3.4 is directly inferred from CCNC measurements. To my understanding, the $\kappa$ value obtained from CCNC data is representative for all particles of equal size, and thus, reflects the possible presence of non-hygroscopic black carbon. This would mean that $\kappa_{median} \rightarrow \kappa_{coating}$ only when the fraction of BC containing particles $\rightarrow 0$.

According to the manuscript BC-free particles "represent majority of the particles" (Page 14, Line 15), and therefore, the definition of $\kappa_{coating} := \kappa_{median}$ would be justified. Is this rationale correct or have I misunderstood the applied notation? In any case, I'd like to ask the authors to describe the reasoning behind $\kappa_{coating} := \kappa_{median}$ more carefully to improve readability and to avoid any danger of misunderstanding.

This leads me to another question: can you quantify "majority of the particles"? For example, would it be useful/possible to have a plot estimating the number or volume fraction of particles with BC core as a function of dry particle size (e.g. in supplementary material)?

**Page 11, Line 15:** The authors state that the anomalies in the size-dependence of $\kappa$ are likely due to the increased uncertainties in CCNC calibration at the lowest and highest supersaturation. In the next two paragraphs, however, the results from these two supersaturations are being discussed more detailed and the authors even use the measured value of $\kappa_{median} = 0.6$ (at SS = 1.33%) to support their hypothesis on nighttime accommodation of ammonium nitrate. Frankly, this would not make much sense if the anomalies in the size dependence of $\kappa$ were solely due to calibration uncertainties. It should be addressed more carefully how the CCNC calibration uncertainties effect the data and data interpretation.

**Page 11, Line 36:** The authors have done great job assessing the contribution of different sources (traffic and wood burning) on the mixing state and presence of non-hygroscopic particles. However, it feels that such a comprehensive analysis and pre-

sentation shifts the attention away from the focal points of the manuscript. I would like to ask the authors to consider condensing this part of the manuscript by moving "less important" parts and maybe some of the figures to the supplementary material and to concentrate especially on those periods relevant for analyzed fog events.

**Page 15, Line 6:** According to Fig. 3, the range between the 95% confidence intervals also illustrates the range of variation during the fog events. Therefore, the derived uncertainty of $SS_{peak}$ (Table 2) could be somewhat interpreted as an indicator of temporal variation. In my opinion, these uncertainty estimates should be discussed, or at the very least, mentioned in this paragraph.

**Page 37, Figure 10:** The figure caption says, "The variability in the fog-activated fraction induced by the choice of $\kappa_{coating}$ (retrieved $\kappa_{median} \pm 0.05$) is represented by horizontal bars". Why is an arbitrary (?) uncertainty of 0.05 used and not the uncertainty indicated by the 95% confidence intervals like in Table 2?

**Technical comments:**

**Page 5, Line 17:** This sentence needs some minor rephrasing as something seems to be lacking, e.g., "...from 20 to 593 nm in 5.5 min, after which the monodisperse aerosol..."

**Page 5, Line 30:** "...was used behind the total inlet..." Should this say interstitial inlet instead of total inlet?

**Page 16, Line 5:** The sentence starting as "The BC cores with..." is not easy to understand and could be rephrased to improve readability.

**Figures:** Is it possible to increase the font sizes especially in Figures 3, 5, 7 and 11.

**References**:

Z. Maalick, T. Kühn, H. Korhonen, H. Kokkola, A. Laaksonen, S. Romakkaniemi, Effect of aerosol concentration and absorbing aerosol on the radiation fog life cycle, Atmospheric Environment, 133, pp. 26-33, https://doi.org/10.1016/j.atmosenv.2016.03.018, 2016.

---

## Referee Comment (RC3) · Anonymous Referee #1 · 30 Oct 2018

This study presents the measurement of BC activation by droplet in real world, the topic is within the scope of ACP. I think there are a few places needing to be addressed before it can be accepted.

Firstly as there is no page number, it is hard to make specific comment. The abstract is too long, I would say maximum 2 paragraphs or better with 1 paragraph. It is recommended to include the previous studies in the introduction on BC heating on clouds, reducing cloud cover, decreasing cloud albedo.

The crucial part of this study is how the scavenging fraction has been measured, some of the technical points need to be more clearly addressed:
a) what is the collection efficiency of the total inlet on collecting droplet, i.e. what is the 50% size cut-off for the droplets, some large droplets may be missed?

b) Will the heating of inlet affect the coating amount of coating compositions of BC.

c) A clear plot is needed to show how the comparison looks between total and interstitial concentration for non-fog period. From the description in the text, this scaling varied from time to time, you may need to show a time series of this scaling ratio, and how this scaling ratio was affecting the results, and why.

d) Also as stated: "For the scanning mobility particle sizer instruments, size-dependent scaling factors were calculated for each fog event in order to take into account both the different line losses behind each inlet and the internal measurement errors of each SMPS." This should be clearly shown by figure.

More explicit definition of internally or externally mixed BC is needed.

Could you also give the scavenging mass fraction of BC or non-BC particles.

What is the black colour in Fig. 11.

A plot showing how the LWC of fog has been associated with SS and related scavenging fraction. What is the source origin of the particles, backtrajectory analysis? A map of the site will help a lot.

How is ACSM used?

"However, Figure 9c clearly shows that droplet activation of BC-containing particles is the mechanism that explains the incorporation of BC cores into fog droplets in the present study: if coagulation between BC particles and fog droplets was giving a dominant contribution, then the fog-activated fraction of BC particles would exhibit much less size and coating dependence and rather with opposite trends." This discussion is not clear at all, so have you observed the coagulation of the BC with droplet? what "opposite trends" are they? ACPD
"Six calibrations were performed, including pre and post campaign, and standard data analysis procedures using the Tofwerk "IgorDAQ" software package (Tofwerk AG, Thun, BE, Switzerland) were applied (reference)." What reference is it?

The key conclusion is to say the model combing ZSR and Kohler theory could well predict the BC activation, but there is no clear plot to show this.

---

## Author Comment (AC2) · 23 Jan 2019

RESPONSES TO THE REFEREES AND CHANGES MADE TO THE MANUSCRIPT.

The authors would like to thank the three referees for their constructive comments which helped to make the paper clearer and easier to understand. This document presents, for each comment from the referees, a response and a note clarifying what has been changed in the manuscript. Indications of page and line numbers refer to the revised version of the manuscript (without track changes).

Answers of the authors to the interactive comment of Anonymous Referee #2 (Referee

**Comment 2)**

**Anonymous review of manuscript: General remarks**

The authors report results from a case study comprising four separate fog events observed in an urban environment in Zurich. Overall, the manuscript is well written and the data analysis has been conducted with great care. The results show that soluble coating on top of an insoluble black carbon (BC) cores indeed increases their ability to serve as condensation nuclei for fog droplets, and the threshold coating thickness decreases with increasing BC core size. Furthermore, the authors demonstrate that a simple -Köhler model can be used to predict the fog droplet activation when the particle size, coating thickness and hygroscopicity of the coating material are known. Understanding the mixing state of ambient BC and its impact and fate in the atmosphere has been of great interest to aerosol community, and thus, the manuscript by Motos et al. is well within the scope of ACP. That said, the main findings of this study are more incremental rather than novel and (as such) provide a little new insight into the studied topic. Therefore, I would like to see more discussion concentrating on the implications of the results, e.g., how black carbon and its aging are currently treated in particle-resolved models (that were also mentioned in the conclusions) and how these new results could possibly improve these aspects. In other words, there is definitely no need to shift the focus of the paper from experimental research into modelling, but instead, highlight the importance of the results and point out more concretely how aerosol community could benefit from them. In my opinion, this would improve the impact of the paper substantially. Otherwise, I only have a few minor comments and suggestions to be considered by the authors.

Response: We thank the referee for the in this article and the suggestions to highlight the potential benefits our main results can bring to the aerosol community. Another paper focusing on the activation of BC in liquid clouds has recently been submitted to ACPD (https://www.atmos-chem-phys-discuss.net/acp-2018-1054/). It combines results from measurements at a high altitude site of clouds with medium to high peak
supersaturation with the results of the present paper of fog with low peak supersaturation. A broader discussion of the activation of BC (in different environments and at different supersaturations) including potential benefits and implications for the modelling community are discussed in more detail in this other paper.

Changes: Here we added the following sentences to Sect. 3.5, p. 19, I. 32: "Several mixing state-resolved modelling studies simulated scavenged fractions based on the estimation of the critical supersaturation using the Köhler theory combined with the ZSR mixing rule (e.g. Matsui, 2016; Ching et al., 2018). The present study suggests that such modelling approaches are valid, at least for fog with low peak supersaturation, and encourages future use of them."

Specific comments from Referee #2:

Comment: "Page 3, Line 21: A relatively recent paper by Maalick et al. (2016) presents results from LEM simulations concentrating on the effect of BC on the evolution and lifetime of radiation fog. Although this specific paper does not directly deal with BC mixing state, it points out an important aspect of BC in aerosol-cloud/fog interactions and could be cited in this paragraph (if the authors wish)."

Response: Agreed by the authors.

Changes: We added the reference to the paragraph mentioned in the comment (p. 3, I. 19): "Although BC can dissipate fog through the semi-direct effect (evaporation of fog droplets due to absorption of solar radiation by BC particles and subsequent droplet evaporation), high concentrations of other CCN were shown to influence fog lifetime in a stronger manner (Maalick et al., 2016). Because these CCN form droplets more efficiently, they lead to increased radiative cooling and decreased droplet removal through sedimentation, thus enhancing fog lifetime."

Comment: "Page 3, Line 35: The study by Dalirian et al. (2018) has been conducted by atomizing BC particles from aqueous solutions and then coating them with organics

**ACPD**
by using a tube furnace. Therefore, it should be referred to as laboratory study rather than a conventional chamber measurement."

Response: We thank the referee for these important details.

Changes: We modified "chamber experiments" by "laboratory studies" in the paragraph mentioned. We also added the following paragraph to Sect. 3.5, p. 19, l. 37: "Dalirian et al. (2018) conducted a laboratory study during which they atomized BC particles from aqueous solutions and then coated them with organics by using a tube furnace."

Comment: "Page 5, Line 26: Later in the paper, the authors are referring to uncertainties in CCN calibration (Sect. 3.1). Therefore, it would be good to briefly describe how the instru-ment was actually calibrated and how the possible instrumental limitations are affecting the measurement uncertainties especially at the lowest and highest supersaturations."

Changes: The following paragraph was added to the experimental section (Sect. 2.2.1), p. 5, l. 32: "The CCNC was calibrated before and after the campaign on 13 August 2015 and 23 March 2016, respectively, using size-selected ammonium sulfate. Both calibration curves agreed within 5% (relative) with each other and are in good agreement with the instrument history for the range between 0.1% and 1.0% SS. This agreement is better than the estimated calibration accuracy of ~10%. As discussed later, the CCNC was also operated at SS = 1.33% during the campaign. Higher uncertainty of  $\pm$ 20% was assigned to this supersaturation to give allowance for extrapolation uncertainty, which may have caused larger bias for data derived from measurements at this SS."

Comment: "Page 8, Line 24: Here, the authors define that the hygroscopicity of the soluble coating  $\kappa$ coating is equal to  $\kappa$ median, which according to Sect. 2.3.4 is directly inferred from CCNC measurements. To my understanding, the value obtained from CCNC data is representative for all particles of equal size, and thus, reflects the possible presence of non-hygroscopic black carbon. This would mean that

**ACPD**
 $\kappa$ median-> $\kappa$ coating only when the fraction of BC containing particles ! According to the manuscript BC-free particles "represent majority of the particles" (Page 14, Line 15), and therefore, the definition of  $\kappa$ coating :=  $\kappa$ median would be justified. Is this rationale correct or have I misunderstood the applied notation? In any case, I'd like to ask the authors to describe the reasoning behind  $\kappa$ coating :=  $\kappa$ median more carefully to improve readability and to avoid any danger of misunderstanding. This leads me to another question: can you quantify "majority of the particles"? For example, would it be useful/possible to have a plot estimating the number or volume fraction of particles with BC core as a function of dry particle size (e.g. in supplemen-tary material)?"

Changes: We added the following paragraph to Sect. 2.3.2 p. 9, l. 16: "[...]We treated our particles as two-component mixtures considering an insoluble BC core ( $\kappa = 0$ ) and a soluble coating to which we assigned the size-resolved median  $\kappa$  value ( $\kappa$ coating:= $\kappa$ median) obtained from sCCNC measurements:  $\kappa$ median was retrieved from the diameter at which 50% activation is reached for a certain SS applied in the CCNC (see Sect. 2.3.4). Figure 7, which will be discussed later, indicates that  $\kappa$ median is virtually not affected by variations in the number fraction of locally emitted BC particles. Instead,  $\kappa$ median is representative of the hygroscopicity of the background aerosol, which has a very small BC mass fraction (e.g: Hueglin et al, 2005), and was therefore chosen as approximation for the coating hygroscopicity. [...]"

Comment: "Page 11, Line 15: The authors state that the anomalies in the sizedependence of  $\kappa$  are likely due to the increased uncertainties in CCNC calibration at the lowest and highest supersaturation. In the next two paragraphs, however, the results from these two supersaturations are being discussed more detailed and the authors even use the measured value of  $\kappa$ median = 0.6 (at SS = 1.33%) to support their hypothesis on night-time accommodation of ammonium nitrate. Frankly, this would not make much sense if the anomalies in the size dependence of were solely due to calibration uncertainties. It should be addressed more carefully how the CCNC calibration uncertainties effect the data and data interpretation.
Response: This apparent confusion is resolved by the fact that the first statement refers to a small deviation, whereas the following two paragraphs refer to substantially higher  $\kappa$ . The text has been modified to avoid this confusion. Moreover, most of the discussion in the two paragraphs is based on temporal patterns, which only relies on precision rather than accuracy of the data.

Changes: First of all, we added uncertainties to the values shown in Table 1. The statement about size dependence of  $\kappa$  was reworded (p. 12, l. 17): "[...] Mean aerosol hygroscopicity increased with increasing particle size (Table 1), a feature which is often observed for atmospheric aerosols (Swietlicki et al., 2008). Note, the aforementioned trend of  $\kappa$  median with particle size is broken for the data from measurements at lowest and highest supersaturations; however, this minor deviation from the trend at either end is likely an artefact caused by systematic bias within the specified calibration uncertainties at these two extreme supersaturations [...]."

We also included a value of uncertainty in the following paragraph, Sect. 3.1, p. 12, l. 38: "The fact that the retrieved  $\kappa$ median value increased up to 0.6 (uncertainty: ±20%) thereby almost reaching the  $\kappa$  value of ammonium nitrate (~0.67 for 0.3%

**(http://fp7.actris.eu/Portals/97/deliverables/PU/WP3\_D3.13\_M24.pdf).**

Comment "Page 11, Line 36: The authors have done great job assessing the contribution of different sources (traffic and wood burning) on the mixing state and presence of non-hygroscopic particles. However, it feels that such a comprehensive analysis and presentation shifts the attention away from the focal points of the manuscript. I would like to ask the authors to consider condensing this part of the manuscript by moving "less important" parts and maybe some of the figures to the supplementary material and to concentrate especially on those periods relevant for analyzed fog events."

Response: Agreed by the authors.

Changes: We moved Figure 5 and the corresponding discussion to the Supplement. We added the following text to Sect. 3.2, p. 13, l. 16 instead: Based on the diurnal cycles of particle and BC concentrations and two different indicators of the source of carbonaceous aerosol (the absorption Ångström exponent and the organics to eBC mass ratio), we conclude that these concentration peaks were caused by traffic emissions, rather than the second most common source of BC in Zurich, wood burning (Zotter et al., 2017; additional discussion attached to Figure S5 in the Supplement).

Comment "Page 15, Line 6: According to Fig. 3, the range between the 95% confidence intervals also illustrates the range of variation during the fog events. Therefore, the derived uncertainty of SSpeak (Table 2) could be somewhat interpreted as an indicator of temporal variation. In my opinion, these uncertainty estimates should be discussed, or at the very least, mentioned in this paragraph.

Response: Indeed, the range between the 95% intervals indeed illustrate temporal variability during a fog event. The authors agree with the reviewer that it should thus not be included in the uncertainty calculation of the mean SSpeak during a fog event. Instead, uncertainties are dominated by extrapolation errors.

Changes: We revised the uncertainty calculations accordingly (see also answer to next
comment) and added the following clarification to Sect 2.3.4, p. 10, l. 35: "As discussed later and shown in Figure 3c,  $\kappa$ median is essentially independent of size for diameters between around 80 nm and 200 nm (between 75 nm and 178 nm for the 14 December event shown in the figure). The uncertainty of  $\kappa$ median extrapolated to the activation cut-off diameters,  $\kappa$ median(D\_half^fog) and  $\kappa$ median(D\_50^fog), is dominated by extrapolation errors, which are estimated to be potentially as large as 40%.

Comment "Page 38, Figure 9: The figure caption says, "The variability in the fogactivated fraction induced by the choice of  $\kappa$ coating (retrieved  $\kappa$ median  $\pm$  0.05) is represented by horizontal bars". Why is an arbitrary (?) uncertainty of 0.05 used and not the uncertainty indicated by the 95% confidence intervals like in Table 2?

Response: The uncertainty analysis for panels b)-e) in Fig. 9 of the revised manuscript was redone. The horizontal error bars now show Poisson-based statistical uncertainties of the activated fractions. We also changed the error bars of SSpeak according to the updated uncertainty estimates (see previous comment). The uncertainty of the  $\kappa$  values, while being important for inferred cloud peak supersaturation, has virtually no influence on the outcome of the closure as changing the  $\kappa$  value has two compensating effects. This is now discussed in detail in the Supplement by means of the new Figure S9 and summarizing statements in the main manuscript.

Changes: Figure 9 (Fig.1 in this author comment) including caption were updated.

The new Figure S9 (Fig.2 in this author comment) in the Supplement and associated discussion were added.

The following summarizing statement was added to Sect. 3.5, p. 18, l. 24: "It is important to note that the closure for the activation of BC-containing particles is insensitive to changes in  $\kappa$  coating as changing  $\kappa$  coating has two compensating effects (see Figure S9 and corresponding discussion in the supplement)."

In addition, the following discussion was attached to Figure S9 in the supplement: "Dis-
cussion of Figure S9: To infer the critical supersaturation of individual BC-containing particles, the hygroscopicity parameter of the coatings,  $\kappa$ coating, was assumed to be equal to the median hygroscopicity measured for the total aerosol ( $\kappa$ median; see Sect. 2.3.2). Here, we performed a sensitivity analysis to test the sensitivity of the BC activation closure result to the assumed value of  $\kappa$ coating: the analysis shown in Figure 9a and 9b and explained in Sect. 3.5 was repeated with using  $\kappa$ coating disturbed by  $\pm 0.05$ . Figure S9b shows that changing  $\kappa$ coating alters the retrieved fog peak supersaturation (solid horizontal lines) as well as the vertical position of the curves indicating the activated fractions. These changes virtually compensate each other such that the observed 50% activated fraction for BC-containing particles is reached at a supersaturation closely mating the fog peak supersaturation for all three  $\kappa$ coating scenarios. This means that successful closure between observed and predicted cloud droplet activation of BC is successfully achieved independent of the exact choice of  $\kappa$ coating."

Technical comments:

Comment "Page 5, Line 17: This sentence needs some minor rephrasing as something seems to be lacking, e.g., "from 20 to 593 nm in 5.5 min, after which the monodisperse aerosol"

Changes: Manuscript corrected.

Comment "Page 5, Line 30: "was used behind the total inlet" Should this say interstitial inlet instead of total inlet?

Changes: Manuscript corrected, we thank the referee.

Comment "Page 16, Line 5: The sentence starting as "The BC cores with" is not easy to understand and could be rephrased to improve readability.

Changes: This sentence (p. 17, l. 13) was changed to: "The BC cores associated to core diameter DrBC below 212 nm and a thin/moderate coating remained smaller than the minimum overall particle diameter required for activation: according to Figure
8, this diameter was around 280 nm during the 14 December event, even for BC-free (water-soluble) particles."

Comment "Figures: Is it possible to increase the font sizes especially in Figures 3, 5, 7 and 11.

Response: Agreed by the authors.

Changes: The changes were implemented in the manuscript.

References (for comments from all referees):

Ching, J., West, M. and Riemer, N.: Quantifying impacts of aerosol mixing state on nucleation-scavenging of black carbon aerosol particles, Atmosphere, 9(1), 17, doi:10.3390/atmos9010017, 2018.

Dalirian, M., Ylisirniö, A., Buchholz, A., Schlesinger, D., Ström, J., Virtanen, A. and Riipinen, I.: Cloud droplet activation of black carbon particles coated with organic compounds of varying solubility, Atmospheric Chemistry and Physics, 18(16), 12477–12489, doi:https://doi.org/10.5194/acp-18-12477-2018, 2018.

Gundel, L. A., Benner, W. H. and Hansen, A. D. A.: Chemical composition of fog water and interstitial aerosol in Berkeley, California, Atmospheric Environment, 28(16), 2715–2725, 1994.

Hallberg, A., Ogren, J. A., Noone, K. J., Heintzenberg, J., Berner, A., Solly, I., Kruisz, C., Reischl, G., Fuzzi, S., Facchini, M. C., Hansson, H.-C., Wiedensohler, A. and Svenningsson, I. B.: Phase partitioning for different aerosol species in fog, Tellus B, 44(5), 545–555, doi:10.1034/j.1600-0889.1992.t01-2-00008.x, 1992.

Hammer, E., Gysel, M., Roberts, G. C., Elias, T., Hofer, J., Hoyle, C. R., Bukowiecki, N., Dupont, J.-C., Burnet, F., Baltensperger, U. and Weingartner, E.: Size-dependent particle activation properties in fog during the ParisFog 2012/13 field campaign, Atmospheric Chemistry and Physics, 14(19), 10517–10533, doi:10.5194/acp-14-10517-

**ACPD**
2014, 2014.

Maalick, Z., Kühn, T., Korhonen, H., Kokkola, H., Laaksonen, A. and Romakkaniemi, S.: Effect of aerosol concentration and absorbing aerosol on the radiation fog life cycle, Atmospheric Environment, 133, 26–33, doi:10.1016/j.atmosenv.2016.03.018, 2016.

Matsui, H.: Black carbon simulations using a size- and mixing-state-resolved threedimensional model: 2. Aging timescale and its impact over East Asia: Size- and Mixing-State-Resolved BC Simulation 2, Journal of Geophysical Research: Atmospheres, 121(4), 1808–1821, doi:10.1002/2015JD023999, 2016.

Nessler, R., Bukowiecki, N., Henning, S., Weingartner, E., Calpini, B. and Baltensperger, U.: Simultaneous dry and ambient measurements of aerosol size distributions at the Jungfraujoch, Tellus B: Chemical and Physical Meteorology, 55(3), 808–819, doi:10.3402/tellusb.v55i3.16371, 2003.

Ohata, S., Moteki, N., Mori, T., Koike, M. and Kondo, Y.: A key process controlling the wet removal of aerosols: new observational evidence, Scientific Reports, 6(1), doi:10.1038/srep34113, 2016.

Petters, M. D. and Kreidenweis, S. M.: A single parameter representation of hygroscopic growth and cloud condensation nucleus activity, Atmospheric Chemistry and Physics, 7(8), 1961–1971, doi:10.5194/acp-7-1961-2007, 2007.

Rose, D., Gunthe, S. S., Mikhailov, E., Frank, G. P., Dusek, U., Andreae, M. O. and Pöschl, U.: Calibration and measurement uncertainties of a continuous-flow cloud condensation nuclei counter (DMT-CCNC): CCN activation of ammonium sulfate and sodium chloride aerosol particles in theory and experiment, Atmospheric Chemistry and Physics, 8(5), 1153–1179, doi:10.5194/acp-8-1153-2008, 2008.

Taylor, J. W., Allan, J. D., Liu, D., Flynn, M., Weber, R., Zhang, X., Lefer, B. L., Grossberg, N., Flynn, J. and Coe, H.: Assessment of the sensitivity of core/shell parameters derived using the single-particle soot photometer to density and refractive index, Atmo-

**ACPD**
spheric Measurement Techniques, 8(4), 1701–1718, doi:10.5194/amt-8-1701-2015, 2015.

Zotter, P., Herich, H., Gysel, M., El-Haddad, I., Zhang, Y., Močnik, G., Hüglin, C., Baltensperger, U., Szidat, S. and Prévôt, A. S. H.: Evaluation of the absorption Ångström exponents for traffic and wood burning in the aethalometer-based source apportionment using radiocarbon measurements of ambient aerosol, Atmospheric Chemistry and Physics, 17(6), 4229–4249, doi:10.5194/acp-17-4229-2017, 2017.

**ACPD**

---

## Author Comment (AC3) · 23 Jan 2019

RESPONSES TO THE REFEREES AND CHANGES MADE TO THE MANUSCRIPT.

The authors would like to thank the three referees for their constructive comments which helped to make the paper clearer and easier to understand. This document presents, for each comment from the referees, a response and a note clarifying what has been changed in the manuscript. Indications of page and line numbers refer to the revised version of the manuscript (without track changes).

Anonymous review of manuscript: General remarks

[Figure]

This paper investigates the activation of internally mixed black carbon in fog by making use of the low supersaturations within fog to do a closure study on the droplet activation behavior of BC-containing particles. The measurements were taken during a field campaign in a residential area of Zurich in the winter, and indicate that aerosols sourced from traffic during rush hour periods are generally less hygroscopic than aerosols sourced from wood burning. The paper is well-written and uses novel methods to demonstrate good agreement between predicated and observed behavior. It is appropriate for ACP and is a useful scientific result that will help to constrain the lifetime of BC in the atmosphere, and demonstrates that simple parameterizations of hygroscopicity in terms of a kappa-Köhler parameter are in good agreement with atmospheric observations. The methods and measurements are adequately described, as are comparisons with previous atmospheric observations. There are a few minor points that should be clar-ified to make the paper clearer. The paper would also benefit from a more focused discussion on the major conclusions of the paper, as it is sometimes challenging to follow.

Specific comments from Referee #3:

Comment: "Some of the figures are hard to read (the text is very small). There are also quite a large number of figures (11) and I would suggest moving some of the less important figures (e.g. figures 4, 5, or 6) to the supplemental information to draw more attention to the other figures."

Changes: We increased the font size in most of the figures. Figure 5 was moved to the Supplement as well as the corresponding text. However, Figure 4 gives a good overview (the only time series) of several parameters during the week of the four fog events and Figure 6 is important to describe the impact of the vehicle emissions during the "rush hours".

Comment: "To improve the clarity of the discussion it would be useful to have a table summarizing the different variables, such as the activation diameters and supersaturations."

Response: The activation diameters, supersaturations, as well as other information concerning the four fog events, are already listed in Table 2.

Changes: We added two references to Table 2 in the manuscript: "Two values of SS-peak are given for each fog event in Table 2" in Sect 3.3 (p. 16, l. 11) and "D_half^fog and D_50^fog lay in the range 320 to 380 nm and 370 to 470 nm, respectively (see Table 2)", also in Sect. 3.3 (p. 16, l. 6).

Comment: "It would be useful to clearly state the upper and lower limits for the optical size range of non-BC containing particles detected by the SP2 in the 8-channel configuration, and at what optical size the scattering detectors are saturated."

Response: Various quantities can be inferred from the data delivered by the 8 channels of the SP2 alone and also from combinations of these data, each of which has its own lower and upper limits and detection and quantification. The dynamic ranges covered by the SP2 for different parameters are directly accessible where needed, i.e. from the range of data shown in the figures: e.g. optical diameter for standard sizing and LEO-fit based sizing in Figs. 3a and 7, rBC mass equivalent core diameter in Fig. 8-10, and coating thickness in dependence of BC core diameter in Fig. 10.

Changes: Several limits of detection are now also explicitly mentioned in the methods section (Sect. 2.2.3). "The respective lower limits of quantification are ~0.32 fg translating to and ~70 nm (note, smaller BC core can also be detected with a detection efficiency of less than unity). At the upper end, BC size distributions are only shown up to 300 nm in diameter, due to insufficient counting statistics at larger sizes." "The peak amplitude of the elastically scattered light is used for optical sizing of BC-free particles from 130 nm to 380 nm."

Comment: "It looks like the laser power in the SP2 used to determine the optical size was only calibrated twice with PSL's, before and after the campaign; were these two

calibrations consistent?"

Changes: One sentence added to the instrumental section (Sect. 2.2.3), p. 6, l. 30: "The laser monitor did not indicate a laser power drift and the calibration coefficient for the scattering detector varied by less than 2% between the two calibrations. Therefore a constant calibration coefficient was applied for the whole campaign."

Comment: "Why was the AMS not used to estimate the index of refraction of the coatings based on the chemical composition of the bulk aerosols? Also, what is the motivation behind choosing the refractive index values for the coatings? These values were given without justification or reference. How much would the index of refraction vary based on the observed bulk aerosol chemical composition, and what is the sensitivity of the calculated kappa values for different values of index of refraction for the BC coating?"

Response: Choosing a refractive index of 1.50 + 0i at 1064 nm very often brings mobility sizing and optical diameter to close agreement for atmospheric aerosols. ACSM derived estimates of the refractive index would not provide additional benefit given the relatively large "representative diameter" of the mass based bulk measurement and the uncertainty of the actual refractive index of the organic fraction at 1064 nm.

Changes: The following addition was made in the methods section (Sect. 2.3.3), p. 6, l. 32:. "Calibrated scattering cross section measurements of BC-free particles were converted to optical diameters (Dopt) assuming spherical particles with a refractive index of 1.50+0i at 1064 nm. With this choice, the particle number size distributions measured by the SMPS and the SP2 agree well in the overlapping size range (not shown) and optical sizing is only weakly sensitive to the applied refractive index (Taylor et al., 2015)."

Comment: "Figure 9 – This size dependence could also potentially be explained by dry deposition removing larger, thickly coated BC particles more efficiently. It would be useful to estimate the relative importance of dry deposition.

Remark: Due to the move of Figure 5 to the Supplement, Figure 9 is now Figure 8. We use the latter name in the paragraph below.

Response: Figure 8c gives an activated fraction, i.e. the fraction of particles that activated to cloud droplets among airborne particles. The brown line in Figure 8c only considers the subset of BC with thick coatings, and gives the activated fraction of this group. Particles deposited to the ground are not considered in this calculation. For example, even if 60% of the thickly coated BC got deposited to the ground by dry processes, the brown line in Figure 8c gives the activated fraction of the other 40% that are still suspended in the air. However, our instrumentation does not allow us to quantify the relative importance of condensation and dry deposition time scales.

Changes: No changes to the manuscript.

Comment: "Also, are there any potential size-dependent biases in using the delay time SP2 method for separating the two populations of aerosols?"

Response: Yes, care needs to be taken with the delay time method because the "delay time" cannot be detected for "thinly coated" small BC cores nor for "thickly coated" large cores. However, we only show BC-core size segregated data and only for the core size range where lower/upper detection limits do not bias the result (see grey shadings in Fig. 8b&c).

Changes: No changes to the manuscript.

References (same for the responses to all referees):

Ching, J., West, M. and Riemer, N.: Quantifying impacts of aerosol mixing state on nucleation-scavenging of black carbon aerosol particles, Atmosphere, 9(1), 17, doi:10.3390/atmos9010017, 2018.

Dalirian, M., Ylisirniö, A., Buchholz, A., Schlesinger, D., Ström, J., Virtanen, A. and Riipinen, I.: Cloud droplet activation of black carbon particles coated with organic compounds of varying solubility, Atmospheric Chemistry and Physics, 18(16), 12477–

12489, doi:https://doi.org/10.5194/acp-18-12477-2018, 2018.

Gundel, L. A., Benner, W. H. and Hansen, A. D. A.: Chemical composition of fog water and interstitial aerosol in Berkeley, California, Atmospheric Environment, 28(16), 2715–2725, 1994.

Hallberg, A., Ogren, J. A., Noone, K. J., Heintzenberg, J., Berner, A., Solly, I., Kruisz, C., Reischl, G., Fuzzi, S., Facchini, M. C., Hansson, H.-C., Wiedensohler, A. and Svenningsson, I. B.: Phase partitioning for different aerosol species in fog, Tellus B, 44(5), 545–555, doi:10.1034/j.1600-0889.1992.t01-2-00008.x, 1992.

Hammer, E., Gysel, M., Roberts, G. C., Elias, T., Hofer, J., Hoyle, C. R., Bukowiecki, N., Dupont, J.-C., Burnet, F., Baltensperger, U. and Weingartner, E.: Size-dependent particle activation properties in fog during the ParisFog 2012/13 field campaign, Atmospheric Chemistry and Physics, 14(19), 10517–10533, doi:10.5194/acp-14-10517-2014, 2014.

Maalick, Z., Kühn, T., Korhonen, H., Kokkola, H., Laaksonen, A. and Romakkaniemi, S.: Effect of aerosol concentration and absorbing aerosol on the radiation fog life cycle, Atmospheric Environment, 133, 26–33, doi:10.1016/j.atmosenv.2016.03.018, 2016.

Matsui, H.: Black carbon simulations using a size- and mixing-state-resolved three-dimensional model: 2. Aging timescale and its impact over East Asia: Size- and Mixing-State-Resolved BC Simulation 2, Journal of Geophysical Research: Atmospheres, 121(4), 1808–1821, doi:10.1002/2015JD023999, 2016.

Nessler, R., Bukowiecki, N., Henning, S., Weingartner, E., Calpini, B. and Baltensperger, U.: Simultaneous dry and ambient measurements of aerosol size distributions at the Jungfraujoch, Tellus B: Chemical and Physical Meteorology, 55(3), 808–819, doi:10.3402/tellusb.v55i3.16371, 2003.

Ohata, S., Moteki, N., Mori, T., Koike, M. and Kondo, Y.: A key process controlling the wet removal of aerosols: new observational evidence, Scientific Reports, 6(1),
* * *
Interactive
comment

doi:10.1038/srep34113, 2016.

Petters, M. D. and Kreidenweis, S. M.: A single parameter representation of hygroscopic growth and cloud condensation nucleus activity, Atmospheric Chemistry and Physics, 7(8), 1961–1971, doi:10.5194/acp-7-1961-2007, 2007.

Rose, D., Gunthe, S. S., Mikhailov, E., Frank, G. P., Dusek, U., Andreae, M. O. and Pöschl, U.: Calibration and measurement uncertainties of a continuous-flow cloud condensation nuclei counter (DMT-CCNC): CCN activation of ammonium sulfate and sodium chloride aerosol particles in theory and experiment, Atmospheric Chemistry and Physics, 8(5), 1153–1179, doi:10.5194/acp-8-1153-2008, 2008.

Taylor, J. W., Allan, J. D., Liu, D., Flynn, M., Weber, R., Zhang, X., Lefer, B. L., Grossberg, N., Flynn, J. and Coe, H.: Assessment of the sensitivity of core/shell parameters derived using the single-particle soot photometer to density and refractive index, Atmospheric Measurement Techniques, 8(4), 1701–1718, doi:10.5194/amt-8-1701-2015, 2015.

Zotter, P., Herich, H., Gysel, M., El-Haddad, I., Zhang, Y., Močnik, G., Hüglin, C., Baltensperger, U., Szidat, S. and Prévôt, A. S. H.: Evaluation of the absorption Ångström exponents for traffic and wood burning in the aethalometer-based source apportionment using radiocarbon measurements of ambient aerosol, Atmospheric Chemistry and Physics, 17(6), 4229–4249, doi:10.5194/acp-17-4229-2017, 2017.
* * *

---

## Author Response (AR1)

**Droplet activation behaviour of atmospheric black carbon particles in fog as a function of their size and mixing state: RESPONSES TO THE REFEREES AND CHANGES MADE TO THE MANUSCRIPT.**

The authors would like to thank the three referees for their constructive comments which helped to make the paper clearer and easier to understand. This document presents, for each comment from the referees, a response and a note clarifying what has been changed in the manuscript. Comments from Referees #1, #2 and #3 can be found Page 2, 7 and 13, respectively. Indications of page and line numbers refer to the revised version of the manuscript (without track changes).

**Answers of the authors to the interactive comment of Anonymous Referee #1 (Referee Comment 3)**

Anonymous review of manuscript: General remarks
"This study presents the measurement of BC activation by droplet in real world, the topic is within the scope of ACP. I think there are a few places needing to be addressed before it can be accepted."

Specific comments from Referee #1:

Comment: "Firstly as there is no page number, it is hard to make specific comment."
*Response*: This issue was already fixed in version2 of the manuscript i.e. as part of the technical correction after the "QuickAccess-Review" stage.

Comment: "The abstract is too long, I would say maximum 2 paragraphs or better with 1 paragraph."
*Response*: Agreed by the authors.
*Changes*: The abstract now reads:

Among the variety of particle types present in the atmosphere, black carbon (BC), emitted by combustion processes, is uniquely associated with harmful effects to the human body and substantial radiative forcing of the Earth. Pure BC is known to be non-hygroscopic, but its ability to acquire a coating of hygroscopic organic and inorganic material leads to increased diameter and hygroscopicity, facilitating droplet activation. This affects BC radiative forcing through aerosol-cloud interactions (aci) and BC life cycle. To gain insights into these processes, we performed a field campaign in winter 2015/16 in a residential area of Zurich which aimed at establishing relations between the size and mixing state of BC particles and their activation to form droplets in fog. This was achieved by operating a CCN counter (CCNC), a scanning mobility particle sizer (SMPS), a single particle soot photometer (SP2) and an aerosol chemical speciation monitor (ACSM) behind a combination of a total- and an interstitial-aerosol inlet.

Our results indicate that in the morning hours of weekdays, the enhanced traffic emissions caused peaks in the number fraction of externally mixed BC particles, which do not act as CCN within the CCNC; compared to nighttime associated to heavily aged internally mixed BC from background air advected to the site. The very low effective peak supersaturations (SSpeak) occurring in fog (between approximately 0.03 and 0.06% during this campaign) restrict droplet activation to a minor fraction of the aerosol burden (around 0.5 to 1% of total particle number concentration between 20 and 593 nm) leading to very selective criteria on diameter and chemical composition. We show that bare BC cores are unable to activate to fog droplets at such low SSpeak, while BC particles surrounded by thick coating have very similar activation behaviour as BC-free particles. Using simplified κ-Köhler theory combined with the ZSR mixing rule assuming spherical core-shell particle geometry constrained with single particle measurements of respective volumes, we found good agreement between the predicted and the directly observed size and mixing state resolved droplet activation behaviour of BC-containing particles in fog. This successful closure demonstrates the predictability of their droplet activation in fog with a simplified theoretical model only requiring size and mixing state information, which can also be applied in a consistent manner in model simulations.

Comment: "It is recommended to include the previous studies in the introduction on BC heating on clouds, reducing cloud cover, decreasing cloud albedo."
*Response*: This is a legitimate request from the referee. Nevertheless, we decided to keep these additions short mainly for two reasons: First, the introduction is already quite comprehensive and our study is quite specific to fog. Second, we have a second manuscript meanwhile submitted to ACPD about BC activation in liquid clouds at a high altitude site (https://www.atmos-chem-phys-discuss.net/acp-2018-1054/), in which such previous studies on the abovementioned BC effects are more directly relevant and therefore also included in the introduction.

*Changes*: We added the following paragraph concerning fog lifetime to the introduction (p. 3, l. 19), as also suggested by Referee #2:

"Although BC can dissipate fog through the semi-direct effect (evaporation of fog droplets due to absorption of solar radiation by BC particles and subsequent droplet evaporation), high concentrations of other CCN were shown to influence fog lifetime in a stronger manner (Maalick et al., 2016). Because these CCN form droplets more efficiently, they lead to increased radiative cooling and decreased droplet removal through sedimentation, thus enhancing fog lifetime."

Comment: "a) what is the collection efficiency of the total inlet on collecting droplet, i.e. what is the 50% size cut-off for the droplets, some large droplets may be missed?"

*Response*: Hammer et al. (2014) use the same inlet for measurements in fog and looked at the influence of sedimentation (SI of their paper). They found that potential systematic bias in the observed activation cut-off diameter remains below 10%.

*Changes*: The following statement has been included on p. 4, l. 28: "Hammer et al. (2014; Supplement) showed that systematic biases in the observed activation cut-off diameter potentially resulting from incomplete collection efficiency of fog droplets in the total inlet remains below 10%".

Comment: "b) Will the heating of inlet affect the coating amount of coating compositions of BC."

*Changes*: The following statement has been included in the experimental section, p. 5, l. 17: "…The temperature increase from outside (~0 °C) to inside (~25 °C) the trailer also contributed to the drying of the sample air and thus evaporation of fog droplet water. Some evaporation artefacts of other semi-volatile aerosol components cannot be excluded. However, they are not expected to be excessive for particles in the upper accumulation mode size range, based on results by Nessler et al. (2003) for comparable temperature difference but at a different location.

Comment: "c) A clear plot is needed to show how the comparison looks between total and interstitial concentration for non-fog period. From the description in the text, this scaling varied from time to time, you may need to show a time series of this scaling ratio, and how this scaling ratio was affecting the results, and why."

*Response*: As mentioned in the manuscript p.5, l. 7, the scaling factor used to correct for line losses in the interstitial inlet was kept at 1.16 for SP2 data before 17 January 2014 and 1.03 afterwards. The initial bias was caused by a pressure drop in the interstitial inlet line, which was fixed on 17 December.

*Changes*: Figure S2a, which shows the effect of line losses on SP2 data and the corrections applied to correct for them, was added to the Supplement.

The reference "see Figure S2a" was added p. 5, l. 10.

[Figure]

**Figure S2: (a) Example SP2-derived particle number size distributions during out-of-cloud conditions showing the corrections made on the interstitial inlet data by the use of scaling factors. A scaling factor of 1.16 was used before 17 December 2015, a factor of 1.03 afterwards.**

Comment: "d) Also as stated: "For the scanning mobility particle sizer instruments, size-dependent scaling factors were calculated for each fog event in order to take into account both the different line losses behind each inlet and the internal measurement errors of each SMPS." This should be clearly shown by figure."

*Changes*: We added Figure S2b to the Supplement, which shows the scaling factor applied to SMPS data for each fog event.

The reference "See Figure S2b" was added p. 5, l. 12.

[Figure]

**Figure S2: (b) Size-dependent scaling factors for correcting SMPS data based on averaged out-of-cloud SMPS measurements before and after each fog event analysed in this study. The replacement of a conductive tubing on 17 December let to a better agreement between the two instruments. The strong size dependence of the scaling factors can be explained by the fact that they originate from two different instruments, the total-inlet and the interstitial-inlet SMPS. For each fog event, the disagreement between both SMPS was rather stable before and after the event, supporting the assumption that this disagreement did not change during the events.**

Comment: "More explicit definition of internally or externally mixed BC is needed".

*Changes*: One sentence added in the introduction (p. 2, l. 14):

"Throughout this study, we refer to BC mixing state in relation to coatings, i.e. a strong degree of internal mixing is associated with thick coatings whereas externally mixed BC is associated with no or very thin coatings."

Comment: "Could you also give the scavenging mass fraction of BC or non-BC particles".

*Changes*: One paragraph added to Sect 3.1, p. 13, l. 1:

"The scavenged mass fractions of BC and the total aerosol, i.e. the mass fraction incorporated into fog droplets, were calculated using the SP2 and the two SMPS assemblies, respectively. The scavenged mass fraction varied between 6% and 12% for BC during the four fog events, and between 15% and 20% for the total aerosol. These results are in close agreement with the fog studies of Hallberg et al. (1992), who reported 6% for elemental carbon (EC) and 18% for sulfate, and somewhat lower than the scavenged fractions of 26% for EC and 38-94% for various inorganic species as reported by Gundel et al. (1994) ."

Comment: "What is the black colour in Fig. 11".

*Response*: This is explained in the caption of Fig. 11:

**"Black pixels in the image indicate 2D-bins for which no particle was found in the total inlet data while at least one particle appeared in the interstitial inlet data, thus leading to a negative fog-activated fraction"**

*Changes*: No changes to the manuscript.

Comment: "A plot showing how the LWC of fog has been associated with SS and related scavenging fraction".

*Response*: LWC is not associated with SS, it was only used to identify fog presence.

*Changes*: We added two sentences in Sect. 2.3.1 (p. 8, l. 23) to clarify this:

"We used a minimum LWC of 100 mg m$^{-3}$ measured by the PVM during at least one hour as threshold to define fog events. Note, the LWC was not used to infer fog peak supersaturation (see Sect. **Error! Reference source not found.**).

Throughout the field campaign, four fog events were retained in the analysis of the present study, all of them between 14 and 20 December 2015 (Table **Error! Reference source not found.**)."

Comment: "What is the source origin of the particles, backtrajectory analysis? A map of the site will help a lot".

*Response*: The diurnal patterns of BC and particle concentrations as well as spectral dependence of aerosol light absorption were used to show the influence of local traffic emissions to the air sampled at the site. This is discussed in Sect. 3.2. Back trajectory analysis would not add further relevant information to this.

*Changes*: We added a map of the measurement site to the Supplement (Figure S1) and references to Figure S1 p. 4, l. 10 and p. 4, l. 15.

[Figure]

**Figure S1: Satellite picture of the Irchel campus and its surroundings. The red cross denotes the location of the measurement site. Map data: ©2018 Google Earth – © 2009 GeoBasis-DE/BKG.**

Comment: "How is ACSM used?"

*Response*: The ACSM was used downstream the total inlet during the campaign. ACSM data were only used for time series of species concentrations in Figure 4.

*Changes*: No changes to the manuscript.

Comment: "However, Figure 9c clearly shows that droplet activation of BC-containing particles is the mechanism that explains the incorporation of BC cores into fog droplets in the present study: if coagulation between BC particles and fog droplets was giving a dominant contribution, then the fog-activated fraction of BC particles would exhibit much less size and coating dependence and rather with opposite trends." This discussion is

not clear at all, so have you observed the coagulation of the BC with droplet? what "opposite trends" are they?"

*Changes*: The following sentences are added to Sect. 3.4, p. 17, l. 31:

"Coagulation scavenging efficiency decreases with increasing particle size, as shown by e.g. Ohata et al. (2016). Therefore, the BC core size and coating thickness dependence of coagulation scavenging would be opposite to the observed relationship. By contrast, the observation is consistent with expectations for nucleation scavenging.

Comment: "Six calibrations were performed, including pre and post campaign, and standard data analysis procedures using the Tofwerk "IgorDAQ" software package (Tofwerk AG, Thun, BE, Switzerland) were applied (reference)." What reference is it?"

*Changes*: This has been corrected.

Comment: "The key conclusion is to say the model combing ZSR and Kohler theory could well predict the BC activation, but there is no clear plot to show this."

*Response*: This is the purpose of Figure 10. And it is also shown in Figure 11.

Detailed explanations are provided in Sect. 3.5 (copied here):

"For each fog event [in Figure 10], 50% fog-activated fraction is reached at an $SS_{crit}$ very close to the $SS_{peak}$ derived from $D_{50}^{fog}$. This agreement confirms that observed activation of BC particles in the fog matches the expected droplet activation behaviour of BC-containing particles as theoretically predicted from independently measured BC-particle properties (size, BC volume fraction and coating hygroscopicity). This demonstrates that closure is successfully achieved, i.e. SP2-based characterization of BC-containing particle properties combined with $\kappa$-Köhler theory is sufficient to accurately describe the activation behaviour of BC-containing particles in fog, despite the fact that either of them are based on the simplifying assumption of spherical core-shell morphology."

And:

"Figure 9b-e also contains the fog-activated fraction of BC-free particles detected by the SP2, for which $SS_{crit}$ was calculated using $\kappa$-Köhler theory with $\kappa_{median}$ and optical diameter from the SP2. 50% activation is by definition reached by those particles with $SS_{crit}$ equal to $SS_{peak}$ inferred from $D_{50}^{fog}$ (small deviations are explained by binning the fog-activated fraction data in supersaturation rather than diameter space). The fact that the activation curves of BC-containing particles in Figure 9b-e agree well with the activation curves of BC-free particles implies the following: the activation of BC-containing particles to fog droplets can be described identical to the activation of BC-free particles but for adjusting the $\kappa$-value with the ZSR-rule to account for the volume fraction of insoluble BC. This is an alternative but equivalent view of how closure is achieved for the activation of BC to fog droplets."

*Changes*: No changes to the manuscript.

**Answers of the authors to the interactive comment of Anonymous Referee #2 (Referee Comment 2)**

Anonymous review of manuscript: General remarks

The authors report results from a case study comprising four separate fog events observed in an urban environment in Zurich. Overall, the manuscript is well written and the data analysis has been conducted with great care. The results show that soluble coating on top of an insoluble black carbon (BC) cores indeed increases their ability to serve as condensation nuclei for fog droplets, and the threshold coating thickness decreases with increasing BC core size. Furthermore, the authors demonstrate that a simple -Köhler model can be used to predict the fog droplet activation when the particle size, coating thickness and hygroscopicity of the coating material are known. Understanding the mixing state of ambient BC and its impact and fate in the atmosphere has been of great interest to aerosol community, and thus, the manuscript by Motos et al. is well within the scope of ACP. That said, the main findings of this study are more incremental rather than novel and (as such) provide a little new insight into the studied topic. Therefore, I would like to see more discussion concentrating on the implications of the results, e.g., how black carbon and its aging are currently treated in particle-resolved models (that were also mentioned in the conclusions) and how these new results could possibly improve these aspects. In other words, there is definitely no need to shift the focus of the paper from experimental research into modelling, but instead, highlight the importance of the results and point out more concretely how aerosol community could benefit from them. In my opinion, this would improve the impact of the paper substantially. Otherwise, I only have a few minor comments and suggestions to be considered by the authors.

*Response*: We thank the referee for the in this article and the suggestions to highlight the potential benefits our main results can bring to the aerosol community. Another paper focusing on the activation of BC in liquid clouds has recently been submitted to ACPD (https://www.atmos-chem-phys-discuss.net/acp-2018-1054/). It combines results from measurements at a high altitude site of clouds with medium to high peak supersaturation with the results of the present paper of fog with low peak supersaturation. A broader discussion of the activation of BC (in different environments and at different supersaturations) including potential benefits and implications for the modelling community are discussed in more detail in this other paper.

*Changes*: Here we added the following sentences to Sect. 3.5, p. 19, l. 32:

"Several mixing state-resolved modelling studies simulated scavenged fractions based on the estimation of the critical supersaturation using the Köhler theory combined with the ZSR mixing rule (e.g. Matsui, 2016; Ching et al., 2018). The present study suggests that such modelling approaches are valid, at least for fog with low peak supersaturation, and encourages future use of them."

Specific comments from Referee #2:

Comment: "**Page 3, Line 21:** A relatively recent paper by Maalick et al. (2016) presents results from LEM simulations concentrating on the effect of BC on the evolution and lifetime of radiation fog. Although this specific paper does not directly deal with BC mixing state, it points out an important aspect of BC in aerosol-cloud/fog interactions and could be cited in this paragraph (if the authors wish)."

*Response*: Agreed by the authors.

*Changes*: We added the reference to the paragraph mentioned in the comment (p. 3, l. 19):

"Although BC can dissipate fog through the semi-direct effect (evaporation of fog droplets due to absorption of solar radiation by BC particles and subsequent droplet evaporation), high concentrations

of other CCN were shown to influence fog lifetime in a stronger manner (Maalick et al., 2016). Because these CCN form droplets more efficiently, they lead to increased radiative cooling and decreased droplet removal through sedimentation, thus enhancing fog lifetime."

Comment: "**Page 3, Line 35:** The study by Dalirian et al. (2018) has been conducted by atomizing BC particles from aqueous solutions and then coating them with organics by using a tube furnace. Therefore, it should be referred to as laboratory study rather than a conventional chamber measurement."
*Response*: We thank the referee for these important details.
*Changes*: We modified "chamber experiments" by "laboratory studies" in the paragraph mentioned.
We also added the following paragraph to Sect. 3.5, p. 19, l. 37:
"Dalirian et al. (2018) conducted a laboratory study during which they atomized BC particles from aqueous solutions and then coated them with organics by using a tube furnace."

Comment: "**Page 5, Line 26:** Later in the paper, the authors are referring to uncertainties in CCN calibration (Sect. 3.1). Therefore, it would be good to briefly describe how the instru-ment was actually calibrated and how the possible instrumental limitations are affecting the measurement uncertainties especially at the lowest and highest supersaturations."
Changes: The following paragraph was added to the experimental section (Sect. 2.2.1), p. 5, l. 32:
"The CCNC was calibrated before and after the campaign on 13 August 2015 and 23 March 2016, respectively, using size-selected ammonium sulfate. Both calibration curves agreed within 5% (relative) with each other and are in good agreement with the instrument history for the range between 0.1% and 1.0% SS. This agreement is better than the estimated calibration accuracy of ~10%. As discussed later, the CCNC was also operated at SS = 1.33% during the campaign. Higher uncertainty of ±20% was assigned to this supersaturation to give allowance for extrapolation uncertainty, which may have caused larger bias for data derived from measurements at this SS."

Comment: "**Page 8, Line 24:** Here, the authors define that the hygroscopicity of the soluble coating $\kappa_{coating}$ is equal to $\kappa_{median}$, which according to Sect. 2.3.4 is directly inferred from CCNC measurements. To my understanding, the value obtained from CCNC data is representative for all particles of equal size, and thus, reflects the possible presence of non-hygroscopic black carbon. This would mean that $\kappa_{median}$->$\kappa_{coating}$ only when the fraction of BC containing particles !
According to the manuscript BC-free particles "represent majority of the particles" (Page 14, Line 15), and therefore, the definition of $\kappa_{coating} := \kappa_{median}$ would be justified. Is this rationale correct or have I misunderstood the applied notation? In any case, I'd like to ask the authors to describe the reasoning behind $\kappa_{coating} := \kappa_{median}$ more carefully to improve readability and to avoid any danger of misunderstanding.
This leads me to another question: can you quantify "majority of the particles"? For example, would it be useful/possible to have a plot estimating the number or volume fraction of particles with BC core as a function of dry particle size (e.g. in supplemen-tary material)?"
Changes: We added the following paragraph to Sect. 2.3.2 p. 9, l. 16:
"[…]We treated our particles as two-component mixtures considering an insoluble BC core ($\kappa = 0$) and a soluble coating to which we assigned the size-resolved median $\kappa$ value ($\kappa_{coating} := \kappa_{median}$) obtained from sCCNC measurements: $\kappa_{median}$ was retrieved from the diameter at which 50% activation is reached for a certain SS applied in the CCNC (see Sect. **Error! Reference source not found.**). Figure 7, which will be discussed later, indicates that $\kappa_{median}$ is virtually not affected by variations in the number fraction of locally emitted BC particles. Instead, $\kappa_{median}$ is representative of the hygroscopicity of the background aerosol, which has a very small BC mass fraction (e.g: Hueglin et al, 2005), and was therefore chosen as approximation for the coating hygroscopicity. […]"

Comment: "**Page 11, Line 15:** The authors state that the anomalies in the size-dependence of $\kappa$ are likely due to the increased uncertainties in CCNC calibration at the lowest and highest supersaturation. In the next two paragraphs, however, the results from these two supersaturations are being discussed more detailed and the authors even use the measured value of $\kappa_{median}$ = 0.6 (at SS = 1.33%) to support their hypothesis on night-time accommodation of ammonium nitrate. Frankly, this would not make much sense if the anomalies in the size dependence of were solely due to calibration uncertainties. It should be addressed more carefully how the CCNC calibration uncertainties effect the data and data interpretation.

*Response*: This apparent confusion is resolved by the fact that the first statement refers to a small deviation, whereas the following two paragraphs refer to substantially higher $\kappa$. The text has been modified to avoid this confusion.

Moreover, most of the discussion in the two paragraphs is based on temporal patterns, which only relies on precision rather than accuracy of the data.

*Changes*: First of all, we added uncertainties to the values shown in Table 1. The statement about size dependence of $\kappa$ was reworded (p. 12, l. 17): "[…] Mean aerosol hygroscopicity increased with increasing particle size (Table 1), a feature which is often observed for atmospheric aerosols (Swietlicki et al., 2008). Note, the aforementioned trend of $\kappa_{median}$ with particle size is broken for the data from measurements at lowest and highest supersaturations; however, this minor deviation from the trend at either end is likely an artefact caused by systematic bias within the specified calibration uncertainties at these two extreme supersaturations […]."

We also included a value of uncertainty in the following paragraph, Sect. 3.1, p. 12, l. 38:
"The fact that the retrieved $\kappa_{median}$ value increased up to 0.6 (uncertainty: ±20%) thereby almost reaching the $\kappa$ value of ammonium nitrate (~0.67 for 0.3% <SS< 1%; Petters and Kreidenweis, 2007), supports this hypothesis."

Concerning the uncertainty of eBC data from the aethalometer, we added the following paragraph p. 7, l. 32:
"The Environmental Technology Verification Report for the Aethalometer reported an instrument precision of ±15% (https://www.epa.gov/etv/pubs/01_vr_aderson_aeth.pdf). However, the uncertainty of aethalometer data, largely dominated by the estimate of the mass-specific attenuation coefficient, can reach values as high as 50%."

Concerning the uncertainty of CCNC data in Table 1 (see Sect. 2.2.1, p. 5, l. 38):
The uncertainties on CCN concentrations measured by the CCNC (Table 1) are based on the study of Rose et al. (2008); they are higher at SS below 0.14%, following the instructions from the ACTRIS standard operation procedures (http://fp7.actris.eu/Portals/97/deliverables/PU/WP3_D3.13_M24.pdf).

Comment "**Page 11, Line 36:** The authors have done great job assessing the contribution of different sources (traffic and wood burning) on the mixing state and presence of non-hygroscopic particles. However, it feels that such a comprehensive analysis and presentation shifts the attention away from the focal points of the manuscript. I would like to ask the authors to consider condensing this part of the manuscript by moving "less important" parts and maybe some of the figures to the supplementary material and to concentrate especially on those periods relevant for analyzed fog events."

*Response*: Agreed by the authors.

Changes: We moved Figure 5 and the corresponding discussion to the Supplement.

We added the following text to Sect. 3.2, p. 13, l. 16 instead:

Based on the diurnal cycles of particle and BC concentrations and two different indicators of the source of carbonaceous aerosol (the absorption Ångström exponent and the organics to eBC mass ratio), we conclude that these concentration peaks were caused by traffic emissions, rather than the second most common source of BC in Zurich, wood burning (Zotter et al., 2017; additional discussion attached to Figure S5 in the Supplement).

Comment **"Page 15, Line 6:** According to Fig. 3, the range between the 95% confidence intervals also illustrates the range of variation during the fog events. Therefore, the derived uncertainty of $SS_{peak}$ (Table 2) could be somewhat interpreted as an indicator of temporal variation. In my opinion, these uncertainty estimates should be discussed, or at the very least, mentioned in this paragraph.

*Response*: Indeed, the range between the 95% intervals indeed illustrate temporal variability during a fog event. The authors agree with the reviewer that it should thus not be included in the uncertainty calculation of the mean $SS_{peak}$ during a fog event. Instead, uncertainties are dominated by extrapolation errors.

*Changes*: We revised the uncertainty calculations accordingly (see also answer to next comment) and added the following clarification to Sect 2.3.4, p. 10, l. 35:

"As discussed later and shown in Figure **Error! Reference source not found.**c, $\kappa_{median}$ is essentially independent of size for diameters between around 80 nm and 200 nm (between 75 nm and 178 nm for the 14 December event shown in the figure). The uncertainty of $\kappa_{median}$ extrapolated to the activation cut-off diameters, $\kappa_{median}(D_{half}^{fog})$ and $\kappa_{median}(D_{50}^{fog})$, is dominated by extrapolation errors, which are estimated to be potentially as large as 40%.

Comment **"Page 38, Figure 9:** The figure caption says, "The variability in the fog-activated fraction induced by the choice of $\kappa_{coating}$ (retrieved $\kappa_{median} \pm 0.05$) is represented by horizontal bars". Why is an arbitrary (?) uncertainty of 0.05 used and not the uncertainty indicated by the 95% confidence intervals like in Table 2?

*Response*: The uncertainty analysis for panels b)-e) in Fig. 9 of the revised manuscript was redone. The horizontal error bars now show Poisson-based statistical uncertainties of the activated fractions. We also changed the error bars of $SS_{peak}$ according to the updated uncertainty estimates (see previous comment). The uncertainty of the $\kappa$ values, while being important for inferred cloud peak supersaturation, has virtually no influence on the outcome of the closure as changing the $\kappa$ value has two compensating effects. This is now discussed in detail in the Supplement by means of the new Figure S9 and summarizing statements in the main manuscript.

*Changes*:

Figure 9 including caption were updated.

[Figure]

**Figure 9: (a):** $SS_{crit}$ **of individual particles sampled behind the total inlet (grey dots) and interstitial inlet (dots coloured by** $\Delta_{coating}$**) as a function of their** $D_{rBC}$ **during the 14 December fog event. The distinct band of data**

points appearing with an $SS_{crit}$ of 0.015 % corresponds to BC-containing particles which caused saturation of the scattering detector even in the leading edge range of the signal, making it impossible to accurately determine $SS_{crit}$. As these particles are known to have lower $SS_{crit}$ than the most thickly coated particles which did not cause signal saturation, they are assigned a "randomly chosen" low value for $SS_{crit}$ and included in the figure. (b), (c), (d), (e): fog-activated fractions of BC-containing (black lines) and BC-free (light blue lines) particles per class of 0.01 % SS for the 14, 15, 18 and 20 December fog events, respectively. The horizontal error bars associated with the activated fractions represent Poisson-based statistical uncertainties. The horizontal blue lines show the $SS_{peak}$ for each fog event retrieved using $D_{50}^{fog}$ (with the method and uncertainty explained in Sect. Error! Reference source not found.).

The new Figure S9 in the Supplement and associated discussion are:

[Figure]

**Figure S9: Sensitivity analysis of BC activated fraction in fog to assumed coating hygroscopicity. Same as Figure 9a and b for the 14 December fog event plus additional activation curves derived with $\kappa_{coating}$ disturbed by ±0.05.**

The following summarizing statement was added to Sect. 3.5, p. 18, l. 24:
"It is important to note that the closure for the activation of BC-containing particles is insensitive to changes in $\kappa_{coating}$ as changing $\kappa_{coating}$ has two compensating effects (see Figure S9 and corresponding discussion in the supplement)."

In addition, the following discussion was attached to Figure S9 in the supplement:
"Discussion of Figure S9: To infer the critical supersaturation of individual BC-containing particles, the hygroscopicity parameter of the coatings, $\kappa_{coating}$, was assumed to be equal to the median hygroscopicity measured for the total aerosol ($\kappa_{median}$; see Sect. 2.3.2). Here, we performed a sensitivity analysis to test the sensitivity of the BC activation closure result to the assumed value of $\kappa_{coating}$: the analysis shown in Figure 9a and 9b and explained in Sect. 3.5 was repeated with using $\kappa_{coating}$ disturbed by ±0.05. Figure S9b shows that changing $\kappa_{coating}$ alters the retrieved fog peak supersaturation (solid horizontal lines) as well as the vertical position of the curves indicating the activated fractions. These changes virtually compensate each other such that the observed 50% activated fraction for BC-containing particles is reached at a supersaturation closely mating the fog peak supersaturation for all three $\kappa_{coating}$ scenarios. This means that successful closure between observed and predicted cloud droplet activation of BC is successfully achieved independent of the exact choice of $\kappa_{coating}$."

**Technical comments:**

Comment **"Page 5, Line 17:** This sentence needs some minor rephrasing as something seems to be lacking, e.g., "from 20 to 593 nm in 5.5 min, after which the monodisperse aerosol"

*Changes*: Manuscript corrected.

Comment **"Page 5, Line 30:** "was used behind the total inlet" Should this say interstitial inlet instead of total inlet?

*Changes*: Manuscript corrected, we thank the referee.

Comment **"Page 16, Line 5:** The sentence starting as "The BC cores with" is not easy to understand and could be rephrased to improve readability.

*Changes*: This sentence (p. 17, l. 13) was changed to:

"The BC cores associated to core diameter $D_{rBC}$ below 212 nm and a thin/moderate coating remained smaller than the minimum overall particle diameter required for activation: according to Figure **Error! Reference source not found.**, this diameter was around 280 nm during the 14 December event, even for BC-free (water-soluble) particles."

Comment **"Figures:** Is it possible to increase the font sizes especially in Figures 3, 5, 7 and 11.

*Response*: Agreed by the authors.

*Changes*: The changes were implemented in the manuscript.

**Answers of the authors to the interactive comment of Anonymous Referee #3 (Referee Comment 1)**

Anonymous review of manuscript: General remarks

This paper investigates the activation of internally mixed black carbon in fog by making use of the low supersaturations within fog to do a closure study on the droplet activation behavior of BC-containing particles. The measurements were taken during a field campaign in a residential area of Zurich in the winter, and indicate that aerosols sourced from traffic during rush hour periods are generally less hygroscopic than aerosols sourced from wood burning.

The paper is well-written and uses novel methods to demonstrate good agreement between predicated and observed behavior. It is appropriate for ACP and is a useful scientific result that will help to constrain the lifetime of BC in the atmosphere, and demonstrates that simple parameterizations of hygroscopicity in terms of a kappa-Köhler parameter are in good agreement with atmospheric observations.

The methods and measurements are adequately described, as are comparisons with previous atmospheric observations. There are a few minor points that should be clarified to make the paper clearer. The paper would also benefit from a more focused discussion on the major conclusions of the paper, as it is sometimes challenging to follow.

Specific comments from Referee #3:

Comment: "Some of the figures are hard to read (the text is very small). There are also quite a large number of figures (11) and I would suggest moving some of the less important figures (e.g. figures 4, 5, or 6) to the supplemental information to draw more attention to the other figures."
*Changes*: We increased the font size in most of the figures. Figure 5 was moved to the Supplement as well as the corresponding text. However, Figure 4 gives a good overview (the only time series) of several parameters during the week of the four fog events and Figure 6 is important to describe the impact of the vehicle emissions during the "rush hours".

Comment: "To improve the clarity of the discussion it would be useful to have a table summarizing the different variables, such as the activation diameters and supersaturations."
*Response*: The activation diameters, supersaturations, as well as other information concerning the four fog events, are already listed in Table 2.
*Changes*: We added two references to Table 2 in the manuscript: "Two values of $SS_{peak}$ are given for each fog event in Table 2" in Sect 3.3 (p. 16, l. 11) and "$D_{half}^{fog}$ and $D_{50}^{fog}$ lay in the range 320 to 380 nm and 370 to 470 nm, respectively (see Table **Error! Reference source not found.**)", also in Sect. 3.3 (p. 16, l. 6).

Comment: "It would be useful to clearly state the upper and lower limits for the optical size range of non-BC containing particles detected by the SP2 in the 8-channel configuration, and at what optical size the scattering detectors are saturated."
*Response*:  Various quantities can be inferred from the data delivered by the 8 channels of the SP2 alone and also from combinations of these data, each of which has its own lower and upper limits and detection and quantification. The dynamic ranges covered by the SP2 for different parameters are directly accessible where needed, i.e. from the range of data shown in the figures: e.g. optical diameter for standard sizing and LEO-fit based sizing in Figs. 3a and 7, rBC mass

equivalent core diameter in Fig. 8-10, and coating thickness in dependence of BC core diameter in Fig. 10.

Changes: Several limits of detection are now also explicitly mentioned in the methods section (Sect. 2.2.3).

"The respective lower limits of quantification are ~0.32 fg translating to and ~70 nm (note, smaller BC core can also be detected with a detection efficiency of less than unity). At the upper end, BC size distributions are only shown up to 300 nm in diameter, due to insufficient counting statistics at larger sizes."

"The peak amplitude of the elastically scattered light is used for optical sizing of BC-free particles from 130 nm to 380 nm."

Comment: "It looks like the laser power in the SP2 used to determine the optical size was only calibrated twice with PSL's, before and after the campaign; were these two calibrations consistent?"

*Changes*: One sentence added to the instrumental section (Sect. 2.2.3), p. 6, l. 30:

"The laser monitor did not indicate a laser power drift and the calibration coefficient for the scattering detector varied by less than 2% between the two calibrations. Therefore a constant calibration coefficient was applied for the whole campaign."

Comment: "Why was the AMS not used to estimate the index of refraction of the coatings based on the chemical composition of the bulk aerosols? Also, what is the motivation behind choosing the refractive index values for the coatings? These values were given without justification or reference. How much would the index of refraction vary based on the observed bulk aerosol chemical composition, and what is the sensitivity of the calculated kappa values for different values of index of refraction for the BC coating?"

*Response*: Choosing a refractive index of $1.50 + 0i$ at 1064 nm very often brings mobility sizing and optical diameter to close agreement for atmospheric aerosols. ACSM derived estimates of the refractive index would not provide additional benefit given the relatively large "representative diameter" of the mass based bulk measurement and the uncertainty of the actual refractive index of the organic fraction at 1064 nm.

*Changes*: The following addition was made in the methods section (Sect. 2.3.3), p. 6, l. 32:.

"Calibrated scattering cross section measurements of BC-free particles were converted to optical diameters ($D_{opt}$) assuming spherical particles with a refractive index of $1.50+0i$ at 1064 nm. With this choice, the particle number size distributions measured by the SMPS and the SP2 agree well in the overlapping size range (not shown) and optical sizing is only weakly sensitive to the applied refractive index (Taylor et al., 2015)."

Comment: "Figure 9 – This size dependence could also potentially be explained by dry deposition removing larger, thickly coated BC particles more efficiently. It would be useful to estimate the relative importance of dry deposition.

*Remark*: Due to the move of Figure 5 to the Supplement, Figure 9 is now Figure 8. We use the latter name in the paragraph below.

*Response*: Figure 8c gives an activated fraction, i.e. the fraction of particles that activated to cloud droplets among airborne particles. The brown line in Figure 8c only considers the subset of BC with thick coatings, and gives the activated fraction of this group. Particles deposited to the ground are not considered in this calculation.

For example, even if 60% of the thickly coated BC got deposited to the ground by dry processes, the brown line in Figure 8c gives the activated fraction of the other 40% that are still suspended in the air.

However, our instrumentation does not allow us to quantify the relative importance of condensation and dry deposition time scales.

*Changes*: No changes to the manuscript.

Comment: "Also, are there any potential size-dependent biases in using the delay time SP2 method for separating the two populations of aerosols?"

*Response*: Yes, care needs to be taken with the delay time method because the "delay time" cannot be detected for "thinly coated" small BC cores nor for "thickly coated" large cores. However, we only show BC-core size segregated data and only for the core size range where lower/upper detection limits do not bias the result (see grey shadings in Fig. 8b&c).
*Changes*: No changes to the manuscript.

References

[revised manuscript text omitted]

**Figure S2: (a) Example SP2-derived particle number size distributions during out-of-cloud conditions showing the corrections made on the interstitial inlet data by the use of scaling factors. These corrections also apply to SMPS data. A scaling factor of 1.16 was used before 17 December 2015, a factor of 1.03 afterwards. (b) Size-dependent scaling factors for correcting SMPS data based on averaged out-of-cloud SMPS measurements before and after each fog event analyzed in this study. The replacement of a conductive tubing on 17 December led to a better agreement between the two instruments. The strong size dependence of the scaling factors can be explained by the fact that they originate from two different instruments, the total-inlet and the interstitial-inlet SMPS. For each fog event, the disagreement between both SMPS was rather stable before and after the event, supporting the assumption that this disagreement did not change during the events.**

[Figure]

**Figure S31:Top: Diurnal variations of mass concentrations measured by the ACSM (top) and corresponding mass fractions (bottom) for the full campaign duration from 6 November 2015 to 31 January 2016. The maximum expected $NH_4^+$ mass concentration, calculated with assuming that all particulate $NO_3^-$ and $SO_4^{2-}$ was neutralized by $NH_4^+$ and that no other anions were present in substantial fraction (i.e. n($NH_4^+$)=2\*n($SO_4^{2-}$)+n($NO_3^-$)), is shown as a dotted line. The measured $NH_4^+$ mass concentration was higher than this maximum calculated concentration; however, the difference is within measurement uncertainty. Mass fractions of organic matter and salts shown in the bottom panel are based on this calculated maximum $NH_4^+$ mass concentration.**

[Figure]

**Figure S4:** Diurnal patterns of the hygroscopicity parameter $\kappa_{median}$ extracted from sCCNC measurement and separately averaged by SS for the whole campaign. In accordance with Table 1, the error bars for SS=1.33 % (purple bars) are also representative of SS=0.14 % (dark red line); the error bars for SS=0.21 % (red line) are representative of all other SS.

[revised manuscript text omitted]

**Figure S85: Activated fractions of the bulk aerosol (SMPS, red lines), BC-containing (SP2, black lines) and BC-free particles (SP2 scattering analysis, light blue line and LEO-fit analysis, green lines) during the 15 (a), 18 (b) and 20 (c) December fog events. The 1-σ uncertainties of the BC-containing particle data are Poisson-based with respect to the BC core number size distribution; the other ones are dominated by the level of (dis-)agreement of the interstitial and total measurements, which was determined during out-of-cloud periods and propagated through the calculation of activated fraction.**

[Figure]

**Figure S9: Sensitivity analysis of BC activated fraction in fog to assumed coating hygroscopicity. Same as Figure 9a and b for the 14 December fog event plus additional activation curves derived with $\kappa_{coating}$ disturbed by ±0.05.**

Discussion of Figure S9: To infer the critical supersaturation of individual BC-containing particles, the hygroscopicity parameter of the coatings, $\kappa_{coating}$, was assumed to be equal to the median hygroscopicity measured for the total aerosol ($\kappa_{median}$; see Sect. 2.3.2). Here, we performed a sensitivity analysis to test the sensitivity of the BC activation closure result to the assumed value of $\kappa_{coating}$: the analysis shown in Figure 9a and 9b and explained in Sect. 3.5 was repeated using $\kappa_{coating}$ disturbed by ±0.05. Figure S9b shows that changing $\kappa_{coating}$ alters the retrieved fog peak supersaturation (solid horizontal lines) as well as the vertical position of the curves indicating the activated fractions. These changes virtually compensate each other such that the observed 50 % activated fraction for BC-containing particles is reached at a supersaturation closely mating the fog peak supersaturation for all three $\kappa_{coating}$ scenarios. This means that successful closure between observed and predicted cloud droplet activation of BC is successfully achieved independent of the exact choice of $\kappa_{coating}$.